# Dual mechanisms of opioid-induced respiratory depression in the inspiratory rhythm-generating network

Nathan A Baertsch[1,2]*, Nicholas E Bush[1], Nicholas J Burgraff[1], Jan-Marino Ramirez[1,2,3]

[1]Center for Integrative Brain Research, Seattle Children's Research Institute, Seattle, United States; [2]Department of Pediatrics, University of Washington, Seattle, United States; [3]Department Neurological Surgery, University of Washington, Seattle, United States

**Abstract** The analgesic utility of opioid-based drugs is limited by the life-threatening risk of respiratory depression. Opioid-induced respiratory depression (OIRD), mediated by the μ-opioid receptor (MOR), is characterized by a pronounced decrease in the frequency and regularity of the inspiratory rhythm, which originates from the medullary preBötzinger Complex (preBötC). To unravel the cellular- and network-level consequences of MOR activation in the preBötC, MOR-expressing neurons were optogenetically identified and manipulated in transgenic mice in vitro and in vivo. Based on these results, a model of OIRD was developed in silico. We conclude that hyperpolarization of MOR-expressing preBötC neurons alone does not phenocopy OIRD. Instead, the effects of MOR activation are twofold: (1) pre-inspiratory spiking is reduced and (2) excitatory synaptic transmission is suppressed, thereby disrupting network-driven rhythmogenesis. These dual mechanisms of opioid action act synergistically to make the normally robust inspiratory rhythm-generating network particularly prone to collapse when challenged with exogenous opioids.

*For correspondence: nathan.baertsch@seattlechildrens.org

**Competing interests:** The authors declare that no competing interests exist.

## Introduction

The neuronal control of breathing is highly vulnerable to exogenous opioid-based analgesics and drugs of abuse. As a result, clinical and illicit use of opioids is associated with the life-threatening, and often difficult to predict, risk for opioid-induced respiratory depression (OIRD) (*Overdyk et al., 2014*; *Gupta et al., 2018*; *Dahan et al., 2018*). Yet, opioids are widely used due to their powerful analgesic utility and their hedonic and addictive properties. In response to overdoses, naloxone (Narcan) remains the gold standard for reversal of OIRD. But naloxone has limitations including a short half-life, loss of analgesia, rapid induction of withdrawal symptoms, and reduced efficacy against opioids with high affinity for the MOR, such as carfentanil and buprenorphine (*Gal, 1989*; *Algera et al., 2019*; *Dahan et al., 2010*; *van Dorp et al., 2006*).

This highlights the need to expand the toolbox of strategies to protect against and reverse OIRD while preserving the intended analgesic effects of opioids. Yet, to date, such strategies are limited. One approach has focused on the development of biased MOR agonists that limit activation of β-arrestin2-dependent signaling (*Conibear and Kelly, 2019*; *Schmid et al., 2017*). However, a role of β-arrestin2 in mediating the respiratory side effects of opioids has not been reproducible among laboratories (*Kliewer et al., 2020*; *Kliewer et al., 2019*; *Bachmutsky et al., 2021*), casting doubt on the potential for biased agonists to mitigate OIRD. A second approach involves the use of respiratory stimulants in combination with opioid medication as a compensatory strategy to protect against OIRD (*Algera et al., 2019*; *Manzke et al., 2003*; *Imam et al., 2020*). Such strategies have shown promise in animal models (*Mosca et al., 2014*; *Kimura et al., 2015*; *Guenther et al., 2010*;

*Ren et al., 2009*; *Sun et al., 2019*; *Haw et al., 2016*; *Dai et al., 2017*) and in some human trials (*Oertel et al., 2010*; *Persson et al., 1999*) but not others (*Lötsch et al., 2005*; *Oertel et al., 2007*). Optimization of this approach will require a detailed mechanistic understanding of the physiological consequences of MOR activation in the respiratory network. Thus, unraveling how opioids affect the respiratory control network represents a critical step toward combating the mortality associated with the opioid health crisis.

Although studying the underlying mechanisms of OIRD in humans remains difficult, in both humans and mice, OIRD is characterized by a pronounced decrease in the frequency and regularity of breaths (*Bouillon et al., 2003*; *Ferguson and Drummond, 2006*; *Smart et al., 2000*). This is primarily due to longer and more irregular pauses between inspiratory efforts (*Drummond, 1983*). Both the beneficial analgesic effects and the detrimental respiratory consequences of opioids are dependent on the Gα$_{i/o}$-coupled, μ-opioid receptor (MOR) encoded by the *Oprm1* gene (*Dahan et al., 2001*; *Sora et al., 1997*). *Oprm1* is expressed widely throughout the brain (*Erbs et al., 2015*) (Allen Brain Atlas), and multiple sites in the central and peripheral nervous system are important for modulating the severity of OIRD (*Montandon et al., 2011*; *Kirby and McQueen, 1986*; *Prkic et al., 2012*; *Mustapic et al., 2010*; *Liu et al., 2021*). Two brainstem sites important for respiratory control – the parabrachial nucleus (PBN) and the preBötzinger complex (preBötC) (*Varga et al., 2020*; *Bachmutsky et al., 2020*) are particularly important for understanding OIRD, since localized genetic deletions of *Oprm1* at these sites abolishes OIRD. The preBötC and PBN are bidirectionally connected (*Yang and Feldman, 2018*; *Yang et al., 2020*) yet have distinct functional roles in the control of breathing (*Baertsch et al., 2018*; *Baertsch and Ramirez, 2019*; *Ramirez and Baertsch, 2018b*). The preBötC is an autonomously rhythmogenic hub for respiratory control, critical for producing the inspiratory rhythm per se (*Del Negro et al., 2018*; *Smith et al., 1991*; *Baertsch et al., 2018*; *Baertsch and Ramirez, 2019*; *Ramirez and Baertsch, 2018b*; *Ramirez et al., 1998*; *Tan et al., 2008*), whereas the PBN is a powerful source of modulatory control, important for providing excitatory drive to the respiratory network and regulating respiratory phase relationships (*Molkov et al., 2013*; *Levitt et al., 2015*; *Zuperku et al., 2017*; *Smith et al., 2013*). Consequently, their respective roles in OIRD may be similarly distinct.

Here, we dissect the network- and cellular-level mechanisms of OIRD within the preBötC. The preBötC contains both excitatory and inhibitory neurons that interact to regulate breathing frequency (*Winter et al., 2009*; *Baertsch et al., 2018*). However, inhibitory synaptic transmission does not seem to play a significant role in OIRD (*Bachmutsky et al., 2020*; *Gray et al., 1999*). Instead, excitatory glutamatergic neurons are the critical substrate for both rhythmogenesis and OIRD in the preBötC (*Funk et al., 1993*; *Greer et al., 1991*; *Bachmutsky et al., 2020*; *Sun et al., 2019*). Collectively, glutamatergic neurons produce an inspiratory rhythm with three distinct time domains. Each respiratory cycle begins in a refractory phase, during which excitability within the preBötC network is reduced and the rhythm is relatively insensitive to perturbations (*Baertsch et al., 2018*; *Kottick and Del Negro, 2015*). The network then transitions to a percolation phase during which excitability gradually builds, driven by intrinsic membrane properties and synaptic excitation among interconnected neurons (*Baertsch and Ramirez, 2019*; *Kam et al., 2013b*). The percolation phase ends when network excitability becomes sufficiently high for interconnected preBötC neurons to produce a synchronized bout of action potentials during the third phase – a network-wide inspiratory burst. Together, the refractory and percolation phases define the time between inspiratory efforts, or interburst interval (IBI), which is the primary determinant of breathing frequency and regularity. Within the preBötC network, each neuron's firing pattern during the respiratory cycle, or 'discharge identity' is largely determined by its synaptic inputs and intrinsic excitability. As a result, excitatory preBötC neurons are not functionally homogeneous (*Ramirez and Baertsch, 2018a*). Indeed, only a subset of preBötC neurons participate in all three phases of the inspiratory rhythm and are therefore considered particularly important mediators of rhythmogenesis (*Kam et al., 2013a*; *Kam et al., 2013b*; *Baertsch and Ramirez, 2019*; *Rubin et al., 2009*). These neurons, referred to as 'pre-inspiratory neurons', are active during inspiratory bursts, suppressed during the refractory phase, and produce spikes during the percolation phase with a characteristic ramp in spike frequency (*Baertsch and Ramirez, 2019*; *Baertsch et al., 2019*). To date, the effects of opioids on preBötC spiking activity and the three-phase inspiratory rhythm have not been well defined, nor have the discharge identities of MOR-expressing preBötC neurons.

To characterize mechanisms underlying OIRD in the preBötC, we combine in vitro and in vivo electrophysiology and computational modeling approaches. Using optogenetic identification and manipulation of *Oprm1* expressing neurons, we find that *Oprm1* is expressed in ~50% of functionally identified preBötC neurons. In the context of OIRD, the activity of *Oprm1+* pre-inspiratory neurons is preferentially suppressed during the percolation phase. However, mimicking this decrease in pre-BötC spiking is not sufficient to phenocopy OIRD. Indeed, we find that, in addition to suppression of the number of spikes produced during the percolation phase, excitatory pre-synaptic transmission from *Oprm1+* neurons is also impaired making the remaining spiking activity of these neurons less consequential for network function. Based on these findings, we developed a computational model of the preBötC containing a subpopulation of *Oprm1+* neurons to isolate and compare the functional consequences of membrane hyperpolarization with reduced pre-synaptic efficacy. Consistent with our electrophysiology results, we find that OIRD is best modeled in silico when both mechanisms occur in combination. We conclude that these dual mechanisms of opioid action in the pre-BötC act together to make the inspiratory rhythm particularly vulnerable to exogenous opioids.

## Results

### Phenotypes of *Oprm1+* preBötC neurons

Neurons in the inspiratory rhythm-generating network are functionally heterogeneous, and the role of any given neuron is determined, in part, by its spiking pattern or 'discharge identity' (*Segers et al., 2012*). The inspiratory network is primarily composed of neurons with four discharge identities that sum to produce the rhythmic activity observed in integrated multi-unit preBötC recordings: (1) *Pre-inspiratory neurons* with spiking during the inter-burst interval that ramps up prior to a bout of action potentials during inspiratory bursts. (2) *Inspiratory neurons* active only during inspiratory bursts. (3) A relatively small number of '*expiratory*' neurons (<15%, *Baertsch et al., 2019*; *Harris et al., 2017*) that receive more inhibitory input than excitatory input during bursts and only spike during the inter-burst interval, and (4) *tonic neurons* with spiking that is not modulated by the inspiratory rhythm. To identify the direct targets of MOR activation in the preBötC, we characterized the discharge identities of *Oprm1* expressing neurons within the preBötC by crossing an *Oprm1-*CreGFP mouse line (referred to as *Oprm1*Cre for short) (*Liu et al., 2021*) with *Rosa26*lsl-ChR2:EYFP (*Rosa26*ChR2 for short) or *Rosa26*lsl-ArchT:EYFP (*Rosa26*ArchT *for short*) mice. Neonatal *Oprm1*Cre; *Rosa26*ChR2 and *Oprm1*Cre; *Rosa26*ArchT offspring were used to produce horizontal brainstem slices (*Anderson et al., 2016*). Single-unit activity (n=223) was recorded simultaneously with rhythmic integrated multi-unit activity from the contralateral preBötC of n=73 slices. The spiking pattern of each unit was referenced to the integrated multi-unit rhythm to determine its discharge identity, and the unit was subsequently characterized optogenetically as *Oprm1+* or *Oprm1−* based on responses to light (*Széll et al., 2020*; *Baertsch et al., 2018*; *Figure 1A*). The *Oprm1* gene was expressed in 52% of pre-inspiratory neurons (n=60), 42% of inspiratory neurons (n=69), 47% of expiratory neurons (n=27), and 60% of tonic neurons (n=67) (*Figure 1B,C*). This leads to the first important conclusion that *Oprm1* is not preferentially expressed among neurons with a particular discharge identity (p=0.803). Of 58 neurons intracellularly labeled with AlexaFluor568, there was no spatial segregation of discharge identities or *Oprm1* expression in the preBötC (*Figure 1D,E*). These results are consistent with previous experiments that utilized in situ hybridization for *Oprm1* transcripts (Allen Brain Atlas; *Figure 1—figure supplement 1*) and imaging of fluorescently tagged MOR (*Erbs et al., 2015*), suggesting that MOR-expressing neurons are numerous and broadly distributed in the pre-BötC region.

### MOR activation reduces spiking during the percolation phase of the inspiratory rhythm

Pre-inspiratory neurons are primarily excitatory (*Baertsch et al., 2019*) and play a critical role in rhythmogenesis (*Del Negro et al., 2018*; *Del Negro et al., 2011*; *Ashhad and Feldman, 2020*), frequency control, and regularity by participating in the percolation phase of the inspiratory rhythm (*Baertsch and Ramirez, 2019*). A prevailing view suggests that excitatory synaptic interactions among pre-inspiratory neurons within the recurrently connected preBötC network play a critical role in the percolation phase and, together with intrinsic membrane properties and synaptic inhibition,

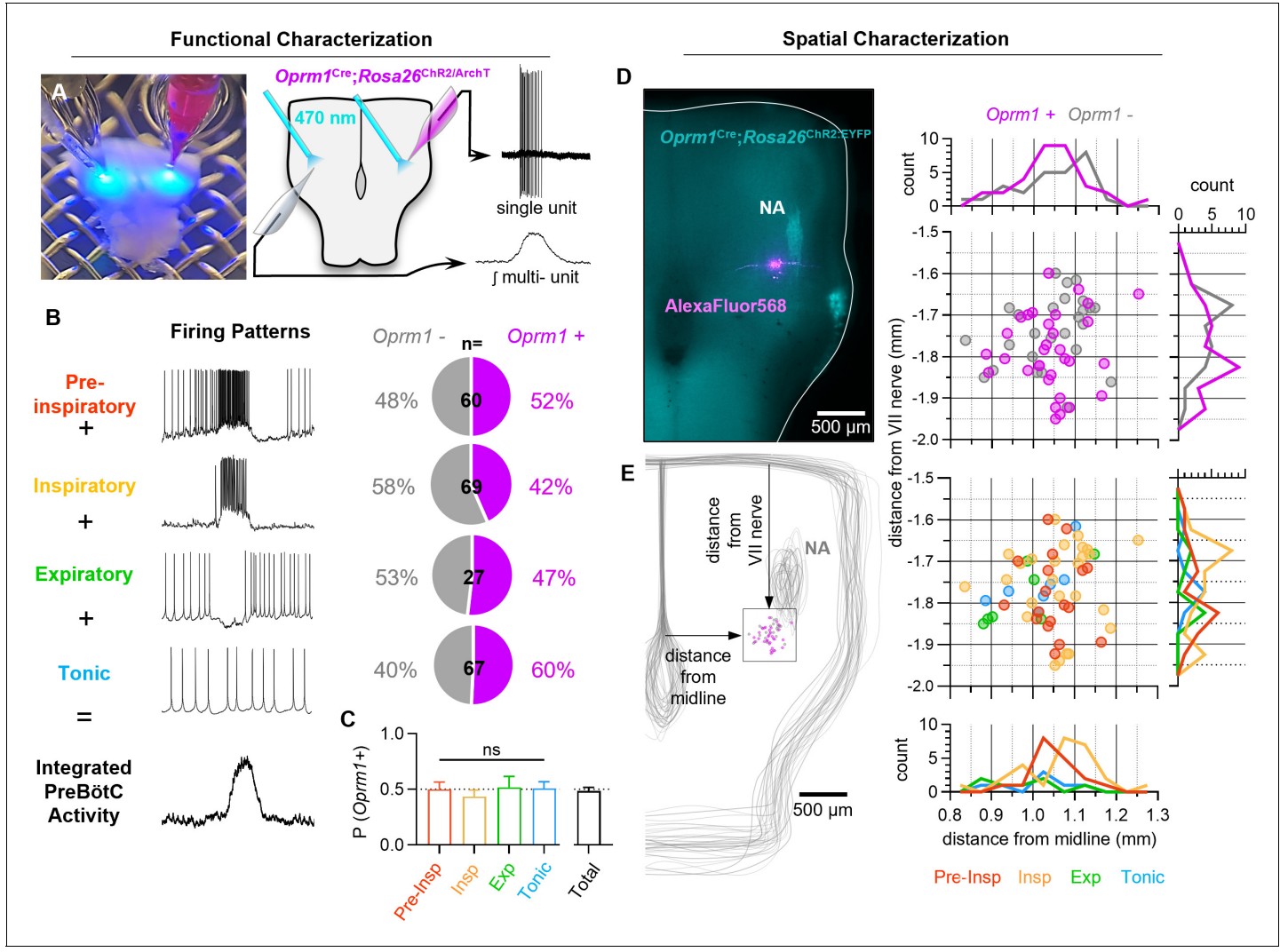

**Figure 1.** Characterization of *Oprm1+* neuron discharge identities and spatial distributions in the preBötC. (**A**) Image (left) and schematic (right) illustrating the approach to optotag functionally identified single units in horizontal brainstem slices. (**B**) Discharge identities of n=223 preBötC neurons (from n=73 horizontal slices) and quantification of *Oprm1* expression among each type demonstrating that (**C**) *Oprm1* is evenly expressed in ~50% of all preBötC neuron types (one-way ANOVA [p=0.82]). (**D**) Image of ChR2:EYFP (cyan) and AlexaFluor568 (magenta) fluorescence in an *Oprm1*^Cre^; *Rosa26*^ChR2^ (Ai32) horizontal brainstem slice following an intracellular recording experiment. Note the enriched *Oprm1−* driven expression of ChR2: EYFP within the nucleus ambiguus (NA). (**E**) Quantified locations of n=58 recorded preBötC neurons from n=35 horizontal slices, caudal and slightly medial to the compact region of the NA, and 2-dimentional distributions in the horizontal plane of *Oprm1+* and *Oprm1−* neurons and pre-inspiratory (pre-insp), inspiratory (insp), expiratory (exp), and tonic neurons. Data shown as means ± SE; ns=not significant.

The online version of this article includes the following source data and figure supplement(s) for figure 1:

**Source data 1.** Characterization of *Oprm1+* neurons.

**Figure supplement 1.** *Oprm1* in situ hybridization experiments #69860840 and #79912572 from the Allen Mouse Brain Atlas (*Lein et al., 2007*) (https://mouse.brain-map.org/search/show?page_num=0&page_size=25&no_paging=false&exact_match=false&search_term=Oprm1&search_type=gene) showing diffuse expression of *Oprm1* gene transcripts within the preBötC region and adjacent Nucleus Ambiguus.

determine the resulting spiking activity (*Kam et al., 2013b*; *Del Negro and Hayes, 2008*; *Del Negro et al., 2011*; *Baertsch and Ramirez, 2019*). To differentiate between the effects of MOR activation on the intrinsically driven versus synaptically driven spiking activity in this important sub-group of preBötC neurons, we examined changes in the spiking of *Oprm1+* and *Oprm1−* pre-inspiratory neurons while increasing the concentration of the MOR agonist DAMGO ([D-Ala2, N-MePhe4, Gly-ol]- enkephalin) before blocking excitatory synaptic transmission (AMPA and NMDA dependent) and reversing MOR activation with Naloxone (5 µM). A single neuron was recoded from each slice

preparation, and all were naïve to opioids at the time of exposure. The blockade of excitatory synaptic inputs revealed that most pre-inspiratory neurons (n=15/20, 75%) continued to spike tonically, i. e., were 'intrinsically tonic', while the others (n=5/20, 25%) became silent, i.e., 'intrinsically quiescent', after blocking excitatory synaptic transmission. To estimate the contribution of intrinsic activity to pre-inspiratory spiking, we normalized the spike frequency during the inter-burst interval to the intrinsic spiking rate after blocking synaptic transmission: intrinsically driven (normalized spike frequency < ~1) versus synaptically driven (normalized spike frequency > ~1) for each individual neuron.

The intrinsic activity of each neuron was predictive of its response to DAMGO. DAMGO reduced ($-60 \pm 13\%$ at 300 nM), but did not eliminate, spiking during the inter-burst interval in *Oprm1+* intrinsically tonic, pre-inspiratory neurons (n=7) (*Figure 2A,C,D*). In contrast, the pre-inspiratory spiking of intrinsically tonic neurons that did not express MOR (*Oprm1−*) was much less affected by DAMGO ($-11 \pm 10\%$ at 300 nM; n=8), an important functional validation of the specificity of the *Oprm1*^Cre mouse line used (*Figure 2A,C,D*). Among intrinsically quiescent neurons (n=5), pre-inspiratory spiking was significantly (p<0.0001) suppressed by DAMGO in both *Oprm1+* and *Oprm1−* neurons ($-95 \pm 5\%$ and $-57 \pm 9\%$ at 300 nM, respectively) (*Figure 2B,E,F*). Collectively, these results suggest that MOR activation suppresses pre-inspiratory percolation activity within the preBötC.

## Effects of MOR activation on preBötC neurons during the burst phase of the inspiratory rhythm

Inspiratory preBötC neurons receive a large volley of concurrent excitatory synaptic drive from multiple input neurons during each burst (*Ashhad and Feldman, 2020*). Single-unit recordings from inspiratory neurons (n=10) that were exclusively active during inspiratory bursts were intrinsically quiescent since they did not spike when deprived of excitatory synaptic input. The spiking frequency of inspiratory neurons during bursts was reduced by DAMGO ($-37 \pm 7\%$ at 300 nM; p<0.0001). Similarly, spiking of pre-inspiratory neurons during bursts was reduced by DAMGO ($-19 \pm 6\%$ at 300 nM; p<0.0001). In contrast to the effects on percolation activity, MOR expression did not predict the changes on burst activity (*Figure 2G,H*), since DAMGO had similar effects on the activity of *Oprm1+* and *Oprm1−* neurons during bursts (pre-inspiratory p=0.187; inspiratory p=0.414). Importantly, these results suggest that any direct effects of MOR activation on the intrinsic excitability of *Oprm1 +* neurons play a minimal role in the suppression of spiking activity during inspiratory bursts.

## Network-level effects of MOR activation in the preBötC

We also explored whether the effects of MOR activation at the cellular level are reflected in corresponding changes at the population level. In horizontal brainstem slices, integrated multi-unit spiking activity was recorded from the preBötC before and during increasing concentrations of DAMGO from 50 to 300 nM. A representative experiment is shown in *Figure 3A*. As expected (*Gray et al., 1999*; *Montandon et al., 2011*; *Wei and Ramirez, 2019*), DAMGO caused a dose-dependent decrease in the frequency of inspiratory bursts, $-26 \pm 3\%$, $-45 \pm 4\%$, $-63 \pm 4\%$, and $-74 \pm 4\%$ change from baseline in 50, 100, 200, and 300 nM DAMGO, respectively (*Figure 3A,C*). Changes in burst frequency did not differ between horizontal slices from heterozygous *Oprm1*^Cre/+ and wild-type controls or between naïve slices and slices previously exposed to DAMGO (*Figure 3—figure supplement 1*). The time between inspiratory bursts became more irregular (based on irregularity scores, see methods) in DAMGO with inconsistent changes in burst-amplitude irregularity (*Figure 3C*). These hallmarks of OIRD were accompanied by a decrease in the total amount of spiking activity between inspiratory bursts, measured as the integrated inter-burst interval (IBI) amplitude (*Figure 3B,C*). Integrated IBI spiking was reduced by $-12 \pm 2\%$, $-15 \pm 2\%$, $-17 \pm 2\%$, and $-18 \pm 2\%$ from baseline in 50, 100, 200, and 300 nM DAMGO, respectively (*Figure 3C*). Changes in burst frequency shared a weak ($R^2=0.16$) but significant linear relationship with changes in IBI spiking at 100 nM DAMGO (p=0.03) and became progressively less significant with 200 nM (p=0.05) and 300 nM (p=0.08) DAMGO (*Figure 3—figure supplement 2*). We also noted that many bursts failed to fully form in DAMGO. These failed bursts were characterized by small-amplitude activity occurring in only subsets of the population typically active during successful bursts (*Figure 3D,E*, *Figure 3—figure supplement 3*), potentially analogous to 'burstlets' (*Kam et al., 2013a*; *Kallurkar et al., 2020*), or mixed-mode oscillations (*Bacak et al., 2016*). As a fraction of the total burst attempts, such burst

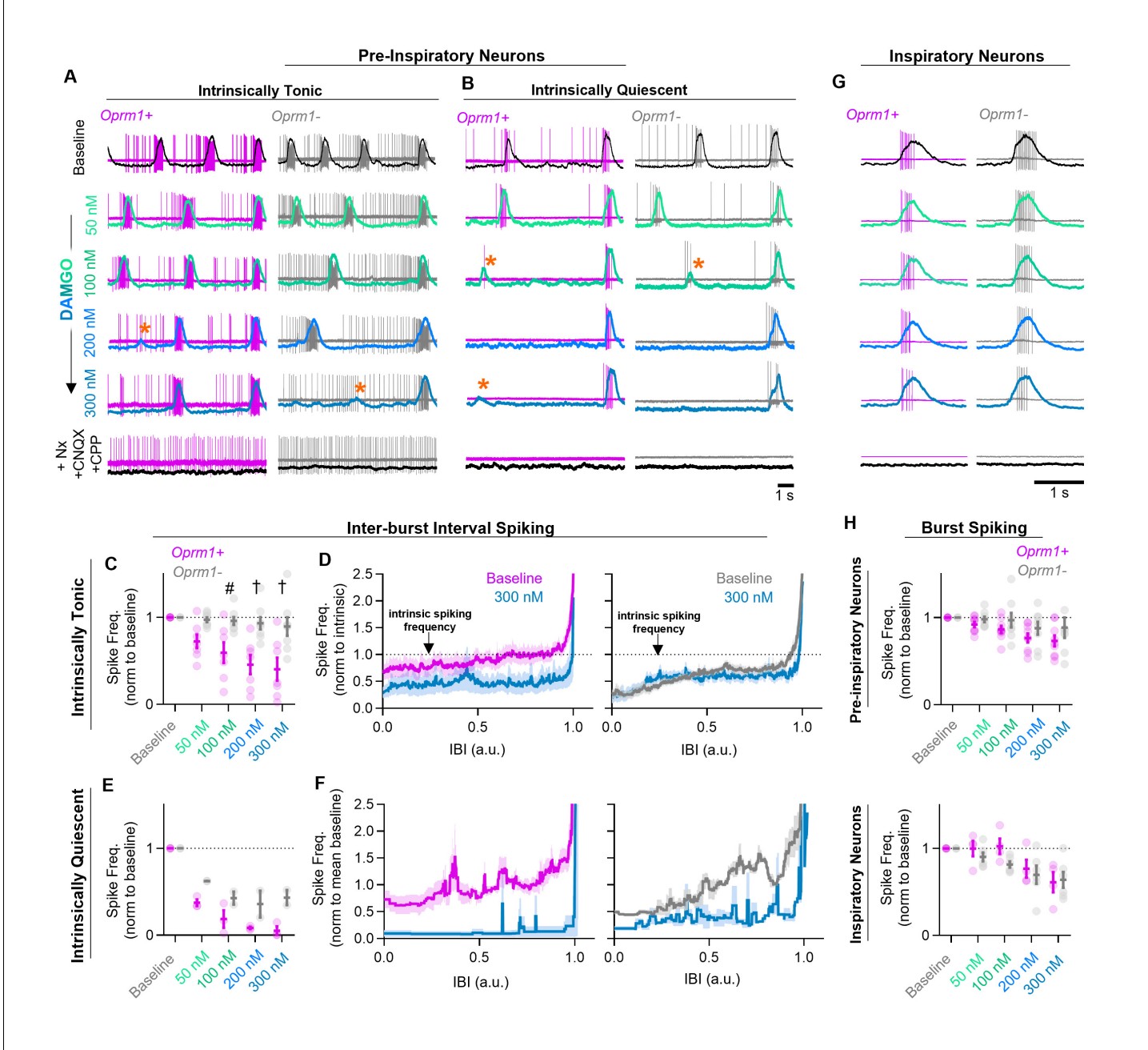

**Figure 2.** Effects of MOR activation on spiking activity of pre-inspiratory and inspiratory preBötC neurons. Example cell-attached recordings from *Oprm1+* and *Oprm1−* intrinsically tonic (**A**) and intrinsically quiescent (**B**) pre-inspiratory neurons during increasing concentrations of DAMGO (50–300 nM) and following subsequent blockade of MOR activation (5 μM Naloxone, Nx) and excitatory synaptic transmission (20 μM CNQX and CPP). Overlayed traces show integrated population activity from the contralateral preBötC. Orange asterisks indicate burst failures. (**C**) Comparison of average IBI spiking frequency from intrinsically tonic *Oprm1+* (n=7) and *Oprm1−* (n=8) pre-inspiratory neurons during increasing concentrations of DAMGO (RM two-way ANOVA [p=0.003] and [Bonferroni post hoc tests]). A single neuron was recorded from each slice preparation. (**D**) Instantaneous spike frequency normalized to intrinsic spiking activity (dotted line) of *Oprm1+* and *Oprm1−* neurons during the IBI at baseline and in 300 nM DAMGO. (**E**) Average IBI spiking frequency from intrinsically quiescent *Oprm1+* (n=3) and *Oprm1−* (n=2) pre-inspiratory neurons during increasing concentrations of DAMGO (RM two-way RM ANOVA [p=0.045] and Bonferroni post hoc tests). (**F**) Instantaneous spike frequency normalized to mean baseline IBI spike rate of *Oprm1+* and *Oprm1−* neurons during the IBI at baseline and in 300 nM DAMGO. (**G**) Example cell-attached recordings from *Oprm1+* and *Oprm1−* inspiratory neurons during increasing concentrations of DAMGO and following subsequent application of Naloxone and blockade of excitatory synaptic transmission. (**H**) Comparison of average spike frequencies during inspiratory bursts (norm to baseline) in *Oprm1+* and *Oprm1−* pre-inspiratory neurons (top) (RM two-way ANOVA, p=0.187) and inspiratory neurons (bottom) (*Oprm1+*, n=4; *Oprm1−*, n=6) (RM two-way ANOVA, p=0.41)

*Figure 2 continued on next page*

Figure 2 continued

during increasing concentrations of DAMGO. Data presented as means± SE; significance of post hoc tests: *p<0.05, #p<0.01, †p<0.001, ‡p<0.0001 compared to baseline.

The online version of this article includes the following source data for figure 2:

**Source data 1.** Changes in neuronal spiking during OIRD.

failures became more prevalent with increasing concentrations of DAMGO (*Figure 3E*) with 43 ± 5% of bursts attempts failing in 300 nM DAMGO.

We also examined the effects of systemic MOR activation on preBötC population activity in vivo. In urethane-anesthetized, adult mice, rhythmic integrated-spiking activity was recorded from the preBötC while simultaneously recording inspiratory motor output from the XII nerve (*Figure 3F*). Example preBötC and XII activity under control conditions (baseline) and ~10 min following intraperitoneal morphine (150 mg/kg) are shown in *Figure 3G*. Changes in total integrated IBI spiking activity before and after morphine are exemplified in *Figure 3H*. Morphine significantly reduced the total preBötC spiking activity during the IBI (IBI spiking) by −17 ± 5%. Breathing frequency was also significantly decreased by morphine (−31 ± 6%) (*Figure 3I*); however, changes in frequency were not related to changes in spiking activity during the IBI (linear regression; p>0.05). At the level of the preBötC, changes in frequency were primarily due to a 74 ± 22% increase in the duration of the IBI from 386 ± 84 ms to 629 ± 97 ms along with a comparatively modest 16 ± 3% increase in the duration of preBötC bursts (TI) (258 ± 14 ms to 300 ± 19 ms) due to slower burst-rise times (*Figure 3J*), consistent with reports of reduced peak inspiratory flow during OIRD (*Ferguson and Drummond, 2006*).

## Hyperpolarization of *Oprm1+* preBötC neurons only partially mimics OIRD

OIRD is often attributed to mechanisms leading to membrane hyperpolarization and reduced spiking activity of respiratory neurons (*Liang et al., 2018*; *Montandon et al., 2016*; *Montandon and Slutsky, 2019*; *Montandon et al., 2011*). However, we found that IBI spiking of *Oprm1+* preBötC neurons was suppressed, but not silenced during OIRD (see *Figure 2*). To address whether this decrease in spiking activity is sufficient to explain the pronounced decrease in inspiratory frequency that occurs when the preBötC network is challenged with exogenous opioids, we employed optogenetic tools and expressed ArchT, an enhanced light activated outward proton pump, specifically within *Oprm1+* neurons. This approach allowed us to examine changes in preBötC network function caused by hyperpolarization of *Oprm1+* neurons.

In horizontal brainstem slices from *Oprm1*^Cre; *Rosa26*^ArchT mice, rhythmic integrated multi-unit preBötC activity was recorded during 10 s continuous bilateral pulses of 598 nm light at 2, 4, and 6 mW. Slices were then exposed to 50, 100, 200, or 300 nM DAMGO, and light-pulse trials were repeated (*Figure 4A–D*). As expected, bilateral hyperpolarization of *Oprm1+* preBötC neurons (*Figure 4—figure supplement 1*) suppressed network spiking during the IBI (−24 ± 3% at 2 mW, −32 ± 4% at 4 mW, and −40 ± 4% at 6 mW). Photoinhibition also depressed the frequency of inspiratory population bursts by −20 ± 7% at 2 mW, −32 ± 10% at 4 mW, and −49 ± 8% at 6 mW, and changes in burst frequency were proportional to the suppression of IBI spiking (burst frequency/IBI spiking ratio: 1.06 ± 0.1 at 2 mW, 1.02 ± 0.16 at 4 mW, and 0.875 ± 0.15 at 6 mW) (*Figure 4C*). Interestingly, subsequent application of DAMGO resulted in a comparatively small change in IBI spiking (−13 ± 2% at 50 nM, −17 ± 3% at 100 nM, −17 ± 3% at 200 nM, and −20 ± 4% at 300 nM) despite a more potent slowing of inspiratory burst frequency (−34 ± 7% at 50 nM, −56 ± 8% at 100 nM, −72 ± 8% at 200 nM, and −83 ± 5% at 300 nM). Indeed, in response to DAMGO, changes in burst frequency were not proportional to changes in IBI spiking (0.74 ± 0.09 at 50 nM, 0.52 ± 0.09 at 100 nM, 0.33 ± 0.10 at 200 nM, and 0.225 ± 0.08 at 300 nM) (*Figure 4C*). Moreover, at 2 mW (but not 4 or 6 mW), bilateral photoinhibition of *Oprm1+* preBötC neurons caused suppressed IBI spiking by an amount equivalent to 300 nM DAMGO (p=0.42; *Figure 4D*), yet inspiratory burst frequency was reduced by only −20 ± 7% during photoinhibiton compared to −83 ± 5% in 300 nM DAMGO (*Figure 4D*). In addition, consistent with our observations from single-unit recordings (see *Figure 2*), MOR activation by DAMGO only partially suppressed spiking activity of *Oprm1+* preBötC neurons.

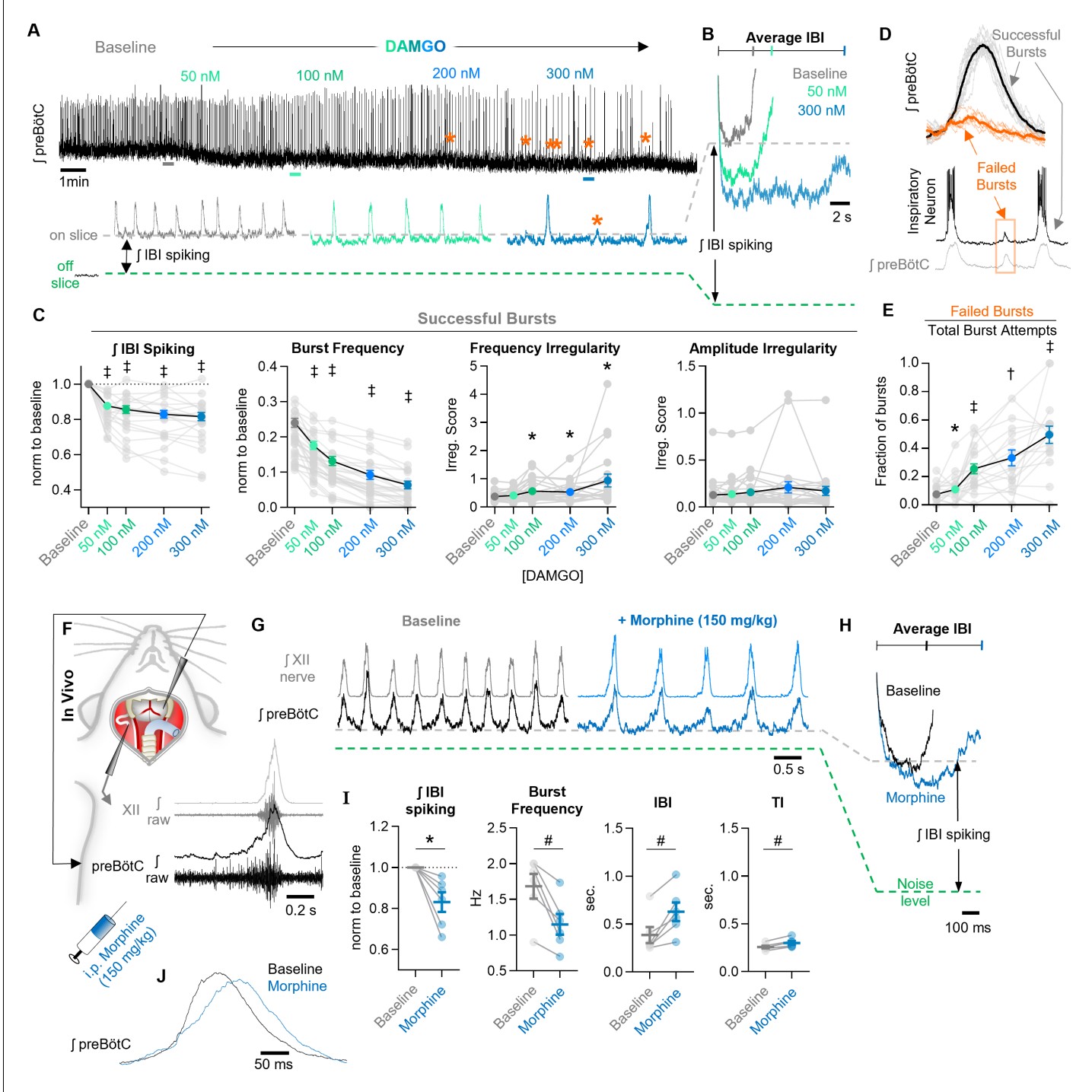

**Figure 3.** Network-level effects of MOR activation on spiking activity in the preBötC in vitro and in vivo. (**A**) Representative ∫preBötC activity from a horizontal slice during increasing concentrations of DAMGO. Orange asterisks indicate burst failures. (**B**) Averaged inter-burst intervals at baseline and in DAMGO demonstrating changes in IBI spiking. (**C**) Quantified ∫IBI spiking (RM one-way ANOVA [p<0.0001] with Bonferroni post hoc tests), burst frequency (RM one-way ANOVA [p<0.0001] with Bonferroni post hoc tests), frequency irregularity (RM mixed-effects analysis [p=0.01] with Dunnett's post hoc tests), and amplitude irregularity (RM mixed-effects analysis [p=0.16] with Dunnett's post hoc tests) from n=30 slices. (**D**) Averaged successful bursts compared to burst failures (top) and example intracellular recording during a failed burst (bottom). (**E**) Quantified fraction of burst failures during increasing concentrations of DAMGO (RM mixed-effects analysis [p<0.0001] with Bonferroni post hoc tests). (**F**) Schematic of in vivo experimental preparation with example simultaneous ∫XII and ∫preBötC recordings. (**G**) Representative ∫XII and ∫preBötC activity at baseline and following i.p. morphine. (**H**) Averaged ∫IBI spiking at baseline and following morphine. (**I**) Quantified ∫IBI spiking, breathing frequency, inter-burst interval, and

*Figure 3 continued on next page*

*Figure 3 continued*

inspiratory time (TI) (n=6; ratio paired t-tests). (J) Example changes in preBötC burst morphology in response to morphine administration in vivo. Data presented as means ± SE; significance of post hoc tests: *p<0.05, #p<0.01, †p<0.001, ‡p<0.0001 compared to baseline.

The online version of this article includes the following source data and figure supplement(s) for figure 3:

**Source data 1.** Network effects of MOR activation.
**Figure supplement 1.** DAMGO dose responses.
**Figure supplement 1—source data 1.** DAMGO dose responses.
**Figure supplement 2.** Relationships between DAMGO-induced changes in inspiratory burst frequency and integrated inter-burst interval spiking in horizontal brainstem slice preparations.
**Figure supplement 2—source data 1.** Network effects of MOR activation.
**Figure supplement 3.** Example whole-cell recording from an inspiratory neuron (top) and corresponding rhythmic multi-unit activity from the contralateral preBötC at baseline and in 300 nM DAMGO.

Even after the inspiratory population rhythm had been nearly silenced by 300 nM DAMGO, light-mediated hyperpolarization of *Oprm1+* preBötC neurons continued to reduce IBI spiking by −39 ± 5% at 6 mW (*Figure 4C*). We conclude that MOR activation causes a significant frequency reduction that is associated with a relatively modest reduction in network-spiking activity in vitro.

We also tested whether optogenetic suppression of preBötC spiking activity is sufficient to mimic OIRD in vivo. PreBötC spiking activity and XII motor output were recorded simultaneously from urethane-anesthetized adult *Oprm1*^Cre; *Rosa26*^ArchT mice during bilateral photoinhibition of *Oprm1+* preBötC neurons (*Figure 4E*). A representative experiment is shown in *Figure 4F*. We found that 6 mW photoinhibition of *Oprm1+* preBötC neurons was sufficient to suppress IBI spiking by −20 ± 5%, an amount similar (p=0.72) to changes in IBI spiking induced by morphine administration (*Figure 4I*). To further examine the cellular consequences of photoinhibition, we recorded single preBötC units in vivo during 6 mW light pulses. We found that photoinhibition reduced, but did not eliminate, spiking from most preBötC neurons (*Figure 4H*). However, breathing frequency was largely unaffected (−1 ± 5% change). Indeed, changes in breathing frequency and the ratio of burst frequency/IBI spiking induced by morphine administration were significantly different than changes observed during photoinhibition of *Oprm1+* preBötC neurons (*Figure 4I*). Thus, our finding that photoinhibition of *Oprm1+* preBötC neurons did not phenocopy OIRD was consistent among in vitro and in vivo preparations, despite their inherent differences for example neonates vs. adults, level of extracellular [K^+], temperature, and effects of anesthesia. Taken together, these results suggest that hyperpolarization and reduced spiking of *Oprm1+* preBötC neurons cannot fully account for OIRD.

## Pre-synaptic drive from *Oprm1+* preBötC neurons is reduced by MOR activation

Based on these observations, we wondered what mechanisms may account for the disproportionate frequency effects of exogenous opioids on preBötC rhythmogenesis relative to their effects on the spiking activity of preBötC neurons. The mechanisms of opioid action are diverse and vary based on brain region (*Crain and Shen, 1990*; *Yaksh, 1997*; *Bourgoin et al., 1994*; *Christie, 1991*). However, in many neuronal circuits including the PBN (*Cramer et al., 2021*) prefrontal cortex (*Yamada et al., 2021*), periaqueductal grey (*Lau et al., 2020*), and hippocampus (*Lu et al., 2021*), opioids have been shown to exert pre-synaptic effects on glutamatergic transmission. Furthermore, in the preBötC, the frequency of miniature EPSCs recorded from individual neurons is reduced by DAMGO (*Wei and Ramirez, 2019*), suggesting that suppression of pre-synaptic transmission may also occur in the respiratory network and contribute to OIRD. To further test this hypothesis, we examined how MOR activation affects evoked EPSPs driven specifically by *Oprm1+* neurons. We first determined the response characteristics of channelrhodopsin2 expressing *Oprm1+* preBötC neurons when directly activated by brief light pulses (10 ms, 0.75 mW, ~50 trials for each neuron). All *Oprm1+* neurons reliably generated a spike (1.13 ± 0.08 spikes/stimulation) with a latency of 9.4 ± 0.6 ms that was consistent among stimulus trials (SD of latency: 1.42 ± 0.22) (*Figure 5A–C*; *Széll et al., 2020*). In contrast, only 7 of 16 (44%) *Oprm1-* neurons produced spikes in response to light pulses. When compared to *Oprm1+* neurons, the spikes generated by *Oprm1−* neurons were unreliable (0.24 ± 0.09 spikes/stimulation) with a longer latency from light onset (19.2 ± 1.6 ms) and exhibited more jitter from trial to trial (SD of latency: 5.8 ± 1.1).

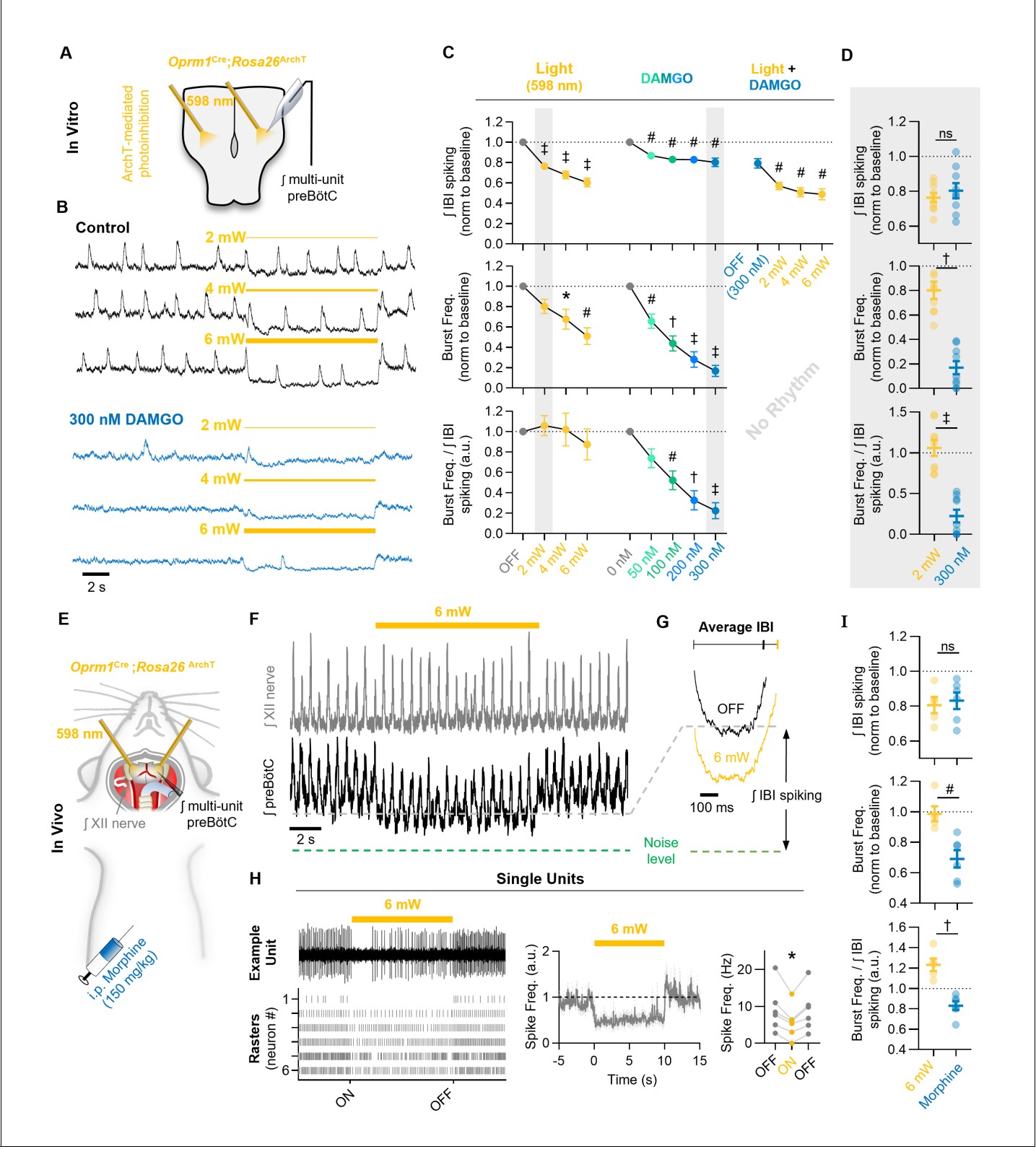

**Figure 4.** Mimicking opioid-induced changes in preBötC spiking does not phenocopy OIRD. (**A**) Schematic of experimental design using *Oprm1*[Cre]; *Rosa26*[ArchT] horizontal brainstem slices. (**B**) Example ∫preBötC recordings during bilateral photoinhbition of preBötC *Oprm1+* neurons at baseline and following OIRD. Note the continued ability to suppress preBötC spiking activity despite silencing of rhythmogenesis with DAMGO. (**C**) Quantification of ∫IBI spiking, burst frequency, and the burst frequency/∫IBI spiking ratio in n=9 slices during baseline photoinhibition of *Oprm1+* preBötC neurons (RM

*Figure 4 continued on next page*

*Figure 4 continued*

one-way ANOVA of ∫IBI spiking [p≤0.0001], RM one-way ANOVA of frequency [p=0.0003], RM one-way ANOVA of frequency/∫IBI spiking ratio [p=0.410]) during increasing concentrations of DAMGO (RM one-way ANOVA of ∫IBI spiking [p<0.0003], RM one-way ANOVA of frequency [p<0.0001], RM one-way ANOVA of frequency/∫IBI spiking ratio [p<0.0001]), and during photoinhibition in 300 nM DAMGO (RM one-way ANOVA of ∫IBI spiking [p=0.0001]). (D) Comparison of changes in ∫IBI spiking, burst frequency, and the burst frequency/IBI spiking ratio during 2 mW photoinhibition and 300 nM DAMGO. Data corresponds to gray highlighted regions in (C) (two-tailed paired t-tests). (E) Schematic of anesthetized in vivo experimental preparation and (F) example ∫XII and ∫preBötC recordings during bilateral photoinhibition of *Oprm1+* preBötC neurons. (G) Average ∫IBI spiking activity at baseline (OFF) and during photoinhibition. (H) Example single unit recording (top left) and rasters from each recording site (bottom left), normalized spike frequency (middle) from n=6 preBötC neurons during a 10 s pulse of 598 nm 6 mW light, and average pre-, during-, and post-light spike frequency (right) (RM one-way ANOVA [p=0.003]). (I) Comparison of changes in ∫IBI spiking, burst frequency, and the burst frequency/IBI spiking ratio elicited during 6 mW bilateral photoinhibition and following i.p. morphine (two-tailed paired t-tests). ns=not significant, *p<0.05, #p<0.01, †p<0.001, ‡p<0.0001 compared to baseline.

The online version of this article includes the following source data and figure supplement(s) for figure 4:

**Source data 1.** Optogenetic hyperpolarization of *Oprm1+* neurons.
**Figure supplement 1.** Optogenetic hyperpolarization of *Oprm1+* preBötC neurons.
**Figure supplement 1—source data 1.** Optogenetic hyperpolarization of *Oprm1+* neurons.

Based on these findings, we tested whether activation of *Oprm1+* neurons would produce corresponding EPSPs in contralateral preBötC neurons via commissural projections, and whether these excitatory interactions are inhibited by MOR activation. Membrane potential was recorded from inspiratory preBötC neurons in whole-cell configuration while delivering light pulses to the contralateral preBötC (*Figure 5D*). Neurons that received excitatory synaptic input from contralateral *Oprm1+* neurons (n=6 of 30 recorded neurons) were then selected based on the presence of consistently evoked EPSPs following the onset of each light pulse. For each of these *Oprm1+* (n=3) and *Oprm1−* (n=3) neurons, evoked EPSPs were recorded during 50–100 stimulus trials under baseline conditions. Stimulus trials were then repeated in 50 nM and 300 nM DAMGO. In some neurons, a hyperpolarizing holding current was applied to maintain $V_m$ below spiking threshold throughout the experiment. The amplitude of EPSPs evoked by activation of contralateral *Oprm1+* neurons was reduced by DAMGO (−20 ± 7% and −49 ± 12% in 50 nM and 300 nM DAMGO, respectively) (*Figure 5E*). Notably, evoked EPSP amplitude was similarly reduced in *Oprm1+* and *Oprm1−* neurons (p=0.6). Collectively, these data indicate that MOR-expressing neurons have commissural projections, and MOR activation suppresses excitatory synaptic transmission from *Oprm1+* neurons to their post-synaptic targets.

## MOR activation limits the ability of *Oprm1+* preBötC neurons to drive the inspiratory rhythm

To test for evidence of impaired synaptic transmission at the population level, integrated multi-unit preBötC activity was recorded during a strong (0.75 mW) sustained (10 s) photoactivation of contralateral *Oprm1+* neurons under baseline conditions and in 300 nM DAMGO. At baseline, photoactivation of *Oprm1+* neurons (*Figure 6—figure supplement 1*) caused a 29 ± 6% increase in the total integrated spiking activity in the contralateral preBötC; however, this effect was significantly reduced to 5 ± 2% in the presence of DAMGO (*Figure 6—figure supplement 2A,B*). The ability of commissural *Oprm1+* neurons to drive an increase in inspiratory burst frequency was also reduced by DAMGO (*Figure 6—figure supplement 1A,C*). Similar results were observed during bilateral photoactivation *Oprm1+* preBötC neurons with 10 s continuous light pulses (three to five trials per light power) (*Figure 6A,B*). Under baseline conditions, light stimulation produced a robust increase in the frequency of inspiratory bursts (53 ± 6% at 0.05 mW, 99 ± 11% at 0.15 mW, and 131 ± 15% at 0.25 mW) (*Figure 6C,D*). *Oprm1+* neuron stimulation also increased network spiking during the IBI (35 ± 4% at 0.05 mW, 81 ± 11% at 0.15 mW, and 95 ± 13% at 0.25 mW). Like the effects of *Oprm1+* neuron hyperpolarization (see *Figure 4*), changes in inspiratory burst frequency were proportional to changes in IBI spiking activity (burst frequency/IBI spiking ratio: 1.13 ± 0.05 at 0.05 mW, 1.12 ± 0.07 at 0.15 mW, and 1.21 ± 0.08 at 0.25 mW) (*Figure 6C,D*). Light pulses were repeated in the presence of 300 nM DAMGO to test how MOR activation may alter the ability of *Oprm1+* neuron depolarization to regulate network function. In 300 nM DAMGO, inspiratory burst frequency was reduced to 37 ± 6% of baseline levels. Burst frequency could be partially restored by depolarization of *Oprm1+*

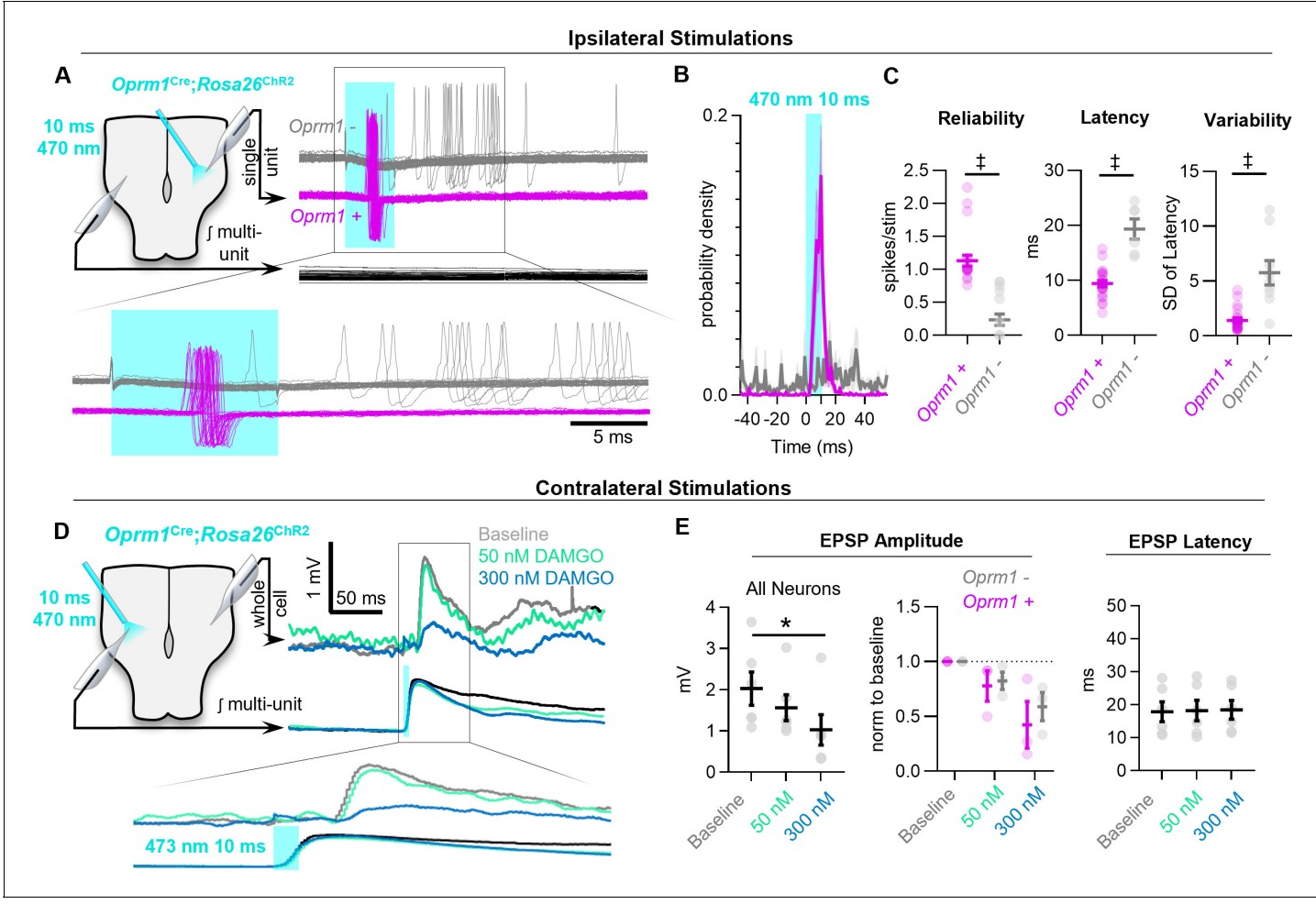

**Figure 5.** MOR activation impairs synaptic transmission in the preBötC. (**A**) Example responses of *Oprm1+* and *Oprm1−* preBötC neurons to direct (ipsilateral) photostimulation. (**B**) Probability density histogram of spikes before, during, and after and 10 ms light pulse (1 ms bins). (**C**) Quantified number of spikes evoked per stimulation, latency to spike from light onset, and variability in spike latencies from n=22 *Oprm1+* and n=6 *Oprm1−* neurons (unpaired two-tailed t-tests). (**D**) Example experiment showing averaged EPSPs evoked during contralateral photostimulation under baseline conditions and in 50 and 300 nM DAMGO (n=6 of 30 neurons exhibited EPSPs during contralateral stimulations). (**E**) Quantified evoked EPSP amplitudes and latencies from n=6 neurons (left: RM one-way ANOVA [p=0.011]; middle: RM two-way ANOVA [p=0.600]; right: RM one-way ANOVA [p=0.511]). Significance of Bonferroni post hoc tests * p<0.05, #p<0.01, †p<0.001, ‡p<0.0001.
The online version of this article includes the following source data for figure 5:

**Source data 1.** MOR activation impairs synaptic transmission.

preBötC neurons to 61 ± 6%, 87 ± 8%, and 95 ± 9% of baseline levels by 0.05, 0.15, and 0.25 mW, respectively (*Figure 6D*). Thus, *Oprm1+* neurons remain functionally integrated within the preBötC network during OIRD. However, the relationship between changes in burst frequency and depolarization of *Oprm1+* neurons, quantified as a slope (Hz/mW), was reduced by DAMGO (0.44 ± 0.05 Hz/mW at baseline vs. 0.21 ± 0.02 Hz/mW in DAMGO; p=0.0004) (*Figure 6E*). In contrast, following MOR activation, changes in the amount of network-spiking activity elicited by *Oprm1+* neuron depolarization (17 ± 12% at 0.05 mW, 57 ± 12% at 0.15 mW, and 77 ± 15% at 0.25 mW) were not different from baseline conditions. Indeed, DAMGO did not alter the relationship between changes in network spiking and depolarization of *Oprm1+* neurons (Slope: 1.32 ± 0.19 IBI spiking/mW at baseline and 1.23 ± 0.15 IBI spiking/mW in 300 nM DAMGO) (*Figure 6E*), suggesting the ability of *Oprm1+* preBötC neurons to spike when depolarized is not impaired by pharmacological MOR activation. The relationship between changes in network spiking and inspiratory frequency remained proportional in DAMGO (burst frequency/IBI spiking ratio: 0.61 ± 0.13 at 0.05 mW, 0.59 ± 0.07 at 0.15 mW, and 0.57 ± 0.06 at 0.25 mW) (*Figure 6D*). However, the relationship was shifted such that larger

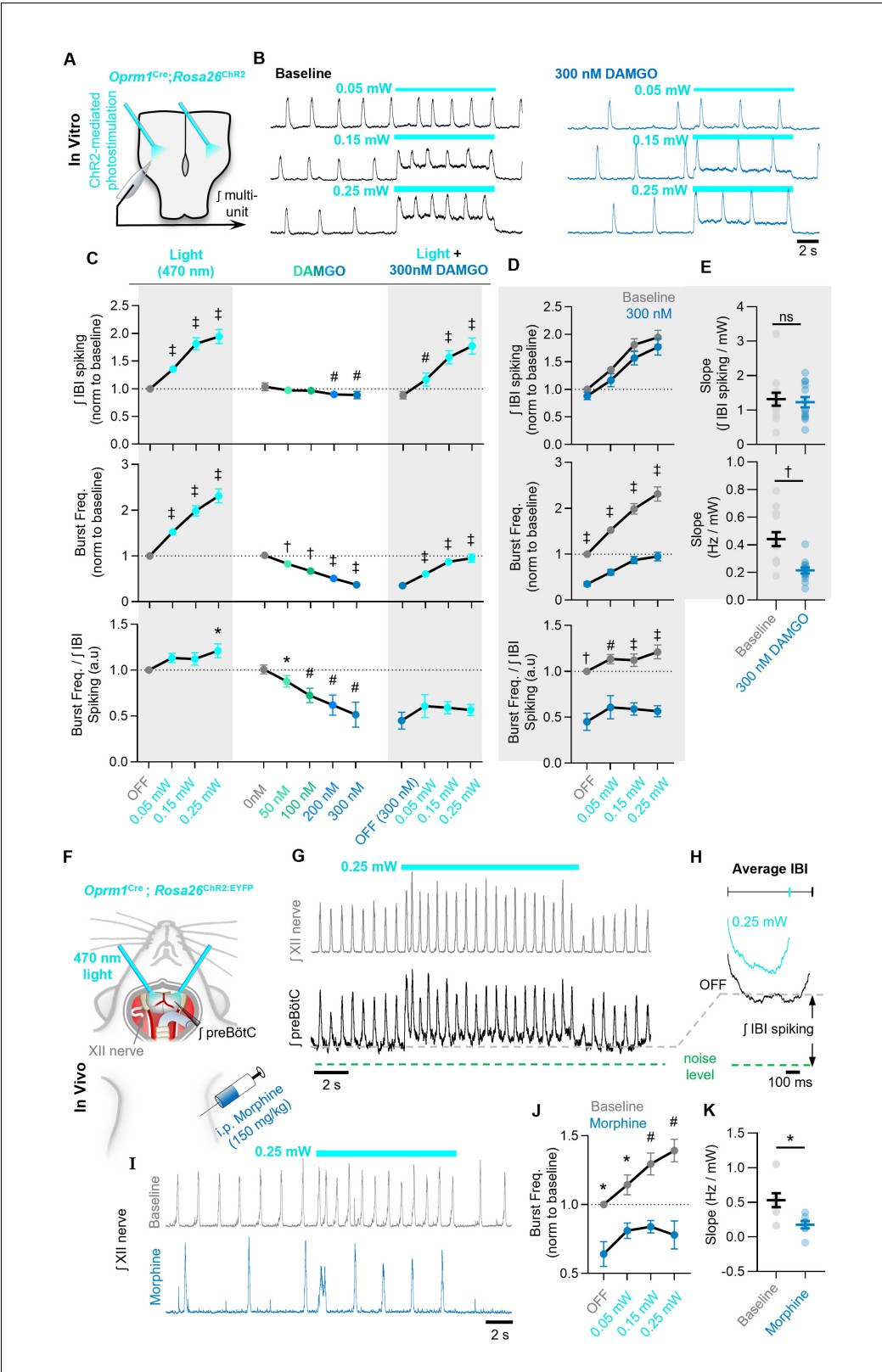

**Figure 6.** The ability of *Oprm1+* neurons to drive preBötC activity is reduced by MOR activation. (A) Experimental schematic and (B) example preBötC activity at baseline and in 300 nM DAMGO during bilateral photostimulation of *Oprm1+* neurons. (C) Quantification of ∫IBI spiking, burst frequency, and the burst frequency/∫IBI spiking ratio in n=13 slices during baseline photostimulation of *Oprm1+* preBötC neurons (RM one-way ANOVA of ∫IBI spiking [p<0.0001], RM one-way ANOVA of frequency [p<0.0001], RM one-way ANOVA of frequency/∫IBI spiking ratio [p=0.038]) during increasing

*Figure 6 continued on next page*

Figure 6 continued

concentrations of DAMGO (RM one-way ANOVA of ∫IBI spiking [p=0.0009], RM one-way ANOVA of frequency [p<0.0001], RM one-way ANOVA of frequency/∫IBI spiking ratio [p=0.003]), and during photostimulation in 300 nM DAMGO (RM one-way ANOVA of ∫IBI spiking [p<0.0001], RM one-way ANOVA of frequency [p<0.0001], RM one-way ANOVA of frequency/∫IBI spiking ratio [p=0.112]). (D) Comparison of light-induced changes in IBI spiking, burst frequency, and the burst frequency/IBI spiking ratio. Data corresponds to gray highlighted regions in (C) (RM two-way ANOVA of ∫IBI spiking [p=0.165], RM two-way ANOVA of frequency [p<0.0001], RM two-way ANOVA of frequency/∫IBI spiking ratio [p<0.0001]). (E) Quantified slope of IBI spiking and burst frequency responses to increasing power of light stimulations (paired two-tailed t-tests). (F) Schematic of in-vivo preparation and (G) representative ∫XII and ∫preBötC activity during 10 s bilateral photostimulation of *Oprm1+* neurons. (H) Averaged ∫IBI activity at baseline (OFF) and during photostimulation. (I) Example inspiratory rhythm (XII) during bilateral photostimulation at baseline and following i.p. morphine. (J) Quantified changes in inspiratory frequency evoked by photostimulation at baseline and after morphine from n=five anesthetized mice (RM two-way ANOVA [p=0.0003]). (K) Quantified slope of burst frequency responses to increasing power of light stimulations (paired two-tailed t-test). Significance of post hoc tests: ns=not significant, *p<0.05, #p<0.01, †p<0.001, ‡p<0.0001.

The online version of this article includes the following source data and figure supplement(s) for figure 6:

**Source data 1.** The ability of *Oprm1+* neurons to drive inspiration is reduced by MOR activation.
**Figure supplement 1.** Optogenetic depolarization of *Oprm1+* preBötC neurons.
**Figure supplement 1—source data 1.** Optogenetic depolarization of *Oprm1+* neurons.
**Figure supplement 2.** MOR activation limits the ability of commissural *Oprm1+* neurons to drive activity in the contralateral preBötC.
**Figure supplement 2—source data 1.** Optogenetic depolarization of *Oprm1+* neurons.
**Figure supplement 3.** Control photostimulations in horizontal slices without ChR2 expression.
**Figure supplement 3—source data 1.** Optogenetic controls.

changes in spiking activity were needed to produce the same change in inspiratory burst frequency. For example, although depolarization of *Oprm1+* preBötC neurons could restore inspiratory burst frequency to ~95% of baseline at the highest light power tested, this required an ~80% increase in network spiking above baseline levels. Consistent with these results in brainstems slices, we found that bilateral photoactivation of *Oprm1+* preBötC neurons in vivo increased IBI spiking (*Figure 6F, G,H*) and produced a light power-dependent increase in breathing frequency (14 ± 7% at 0.05 mW, 29 ± 8% at 0.15 mW, and 39 ± 8% at 0.25 mW). Following morphine administration (150 mg/kg i.p.), photoactivation of *Oprm1+* preBötC neurons continued to increase breathing frequency (*Figure 6I, J*), but to a lesser extent and with less light power-dependence (0.53 ± 0.1 Hz/mW at baseline vs. 0.18 ± 0.05 Hz/mW in DAMGO) (*Figure 6J,K*). In control experiments, preparations lacking opsin expression had no response to preBötC photostimulations (*Figure 6—figure supplement 3*). Collectively, these data suggest that MOR activation causes the spiking activity of *Oprm1+* preBötC neurons to become less consequential for preBötC network function.

## Modeling the functional consequences of preBötC MOR activation in silico

Based on the experimental results in vitro and in vivo described above, we constructed a computational network in silico to model the effects of MOR activation on preBötC rhythmogenesis. Our model network, based on elements from prior preBötC computational studies (*Butera et al., 1999a*; *Butera et al., 1999b*; *Harris et al., 2017*; *Park and Rubin, 2013*), contains 300 total model neurons, 80% designated as excitatory and 20% inhibitory, connected randomly with an average of 6 connections/neuron (*Supplementary file 1*). The intrinsic spiking activity (i.e., without synaptic interactions) of the model neurons was set such that 65% were quiescent (Q) and 35% exhibited tonic spiking (T). No model neurons were defined to be endogenously bursting (i.e., burst in the absence of synaptic inputs). The Butera-type model was chosen to evaluate the potential interactions between hyperpolarization and synaptic suppression on network-level properties as it is computationally tractable, low-dimensional, and known to be capable of incorporating critical inhibitory populations while exhibiting robust bursting dynamics (*Harris et al., 2017*). Specific conductance parameters for each model neuron type are summarized in *Supplementary file 2* and described in Materials and methods. Parameters were taken from *Harris et al., 2017*, and $g_{NaP}$ and $g_{leak}$ were modified to qualitatively match burst frequencies and durations produced in vitro. Based on our optogenetic tagging experiments in vitro (see *Figure 1*), we introduced the parameters $I_{opioid}$ and $syn_{opioid}$ to a subpopulation of *Oprm1+* neurons, encompassing 50% of all excitatory model neurons. $I_{opioid}$ introduces a hyperpolarizing current to the subpopulation of *Oprm1+* model neurons

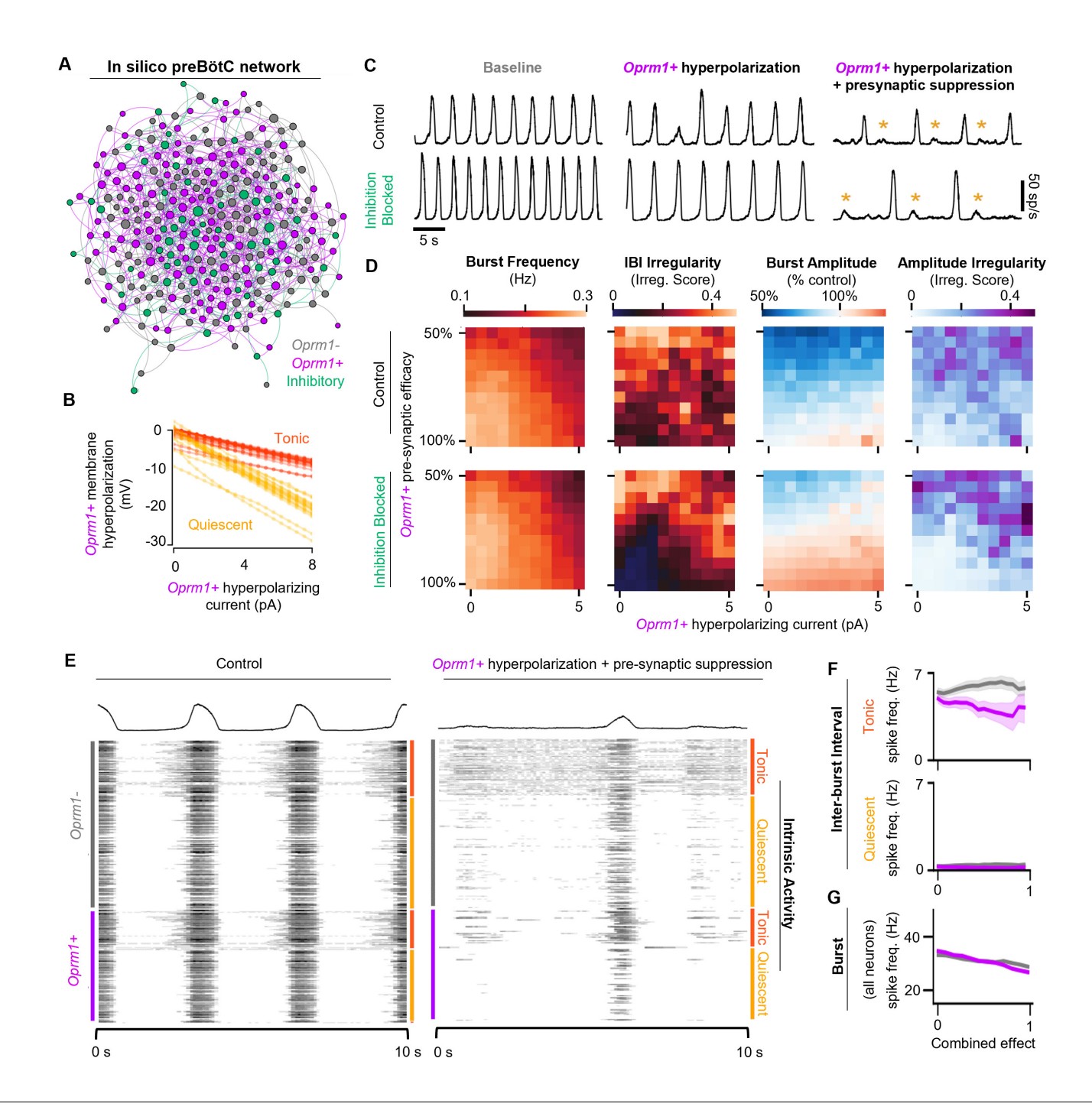

**Figure 7.** Modeling preBötC MOR activation in silico. (A) Example model network structure. Model networks were connected randomly and contained excitatory (80%) and inhibitory neurons (20%) that were intrinsically tonic (35%) or quiescent (65%). Half of all excitatory neurons were designated as *Oprm1+*. Node size and centrality represent the number of synaptic connections. (B) Current/voltage plots for intrinsically tonic (n=17) and quiescent (n=31) *Oprm1+* model neurons for the applied opioid-driven hyperpolarization. (C) Example rhythmic activity from the network shown in (A) at baseline, during 6 pA hyperpolarization of *Oprm1+* neurons, and during 6 pA hyperpolarization and a 45% reduction in pre-synaptic efficacy of *Oprm1+* neurons. Orange asterisks indicate burst failures. (D) Heat maps showing the isolated and combined effects of *Oprm1+* neuron hyperpolarization and pre-synaptic suppression on characteristics of the network rhythm. (E) Example spiking activity of all 300 neurons in the model network at baseline and during simulated MOR activation (6 pA hyperpolarization and 45% pre-synaptic suppression of *Oprm1+* neurons). (F) Quantified inter-burst interval (IBI) spike frequencies of intrinsically tonic and quiescent *Oprm1+* and *Oprm1−* neurons during simulated MOR activation. (G) Spike frequency during

*Figure 7 continued*

network bursts for all *Oprm1+* and *Oprm1-* model neurons during simulated MOR activation (combined effect of 1 designates 6 pA hyperpolarization and 55% pre-synaptic efficacy of *Oprm1+* neurons). Data shown as mean± SE.

The online version of this article includes the following figure supplement(s) for figure 7:

**Figure supplement 1.** A simulated data-driven *Oprm1+* subpopulation is necessary for rhythmogenesis in an in silico preBötC model network.

**Figure supplement 2.** OIRD in the preBötC is best modeled by concurrent hyperpolarization and pre-synaptic suppression of a simulated *Oprm1+* subpopulation.

(*Figure 7B*), whereas $syn_{opioid}$ reduces the strength of synaptic output (i.e., pre-synaptic) from *Oprm1 +* neurons. With these parameters set to 0 (i.e., under control conditions), the model preBötC network produced robust rhythmic bursting activity like preBötC rhythms in vitro (*Figure 7C*).

To test the functional role of the modeled *Oprm1+* subpopulation, *Oprm1+* neurons were removed from the rhythmogenic process by either (1) increasing $I_{opioid}$ such that spikes were no longer generated by *Oprm1+* neurons or (2) increasing $syn_{opioid}$ such that the spikes produced by *Oprm1+* neurons were inconsequential for their post-synaptic targets (*Figure 7—figure supplement 1*). Consistent with the critical role of *Oprm1+* neurons for preBötC rhythmogenesis in vitro (*Bachmutsky et al., 2020*; *Gray et al., 1999*; *Montandon et al., 2011*; *Wei and Ramirez, 2019*; *Mellen et al., 2003*), both methods of functionally removing the modeled *Oprm1+* subpopulation effectively silenced the network rhythm. It is noteworthy, however, that hyperpolarizing *Oprm1+* model neurons vs. blocking their synaptic output had distinct effects on spiking activity generated by the network, reminiscent of the differential effects on network spiking we observed during optogenetic hyperpolarization of *Oprm1+* neurons vs. MOR activation (see *Figure 4*).

We utilized our computational model network to dissociate the functional consequences of the intrinsic vs. synaptic effects of preBötC MOR activation by manipulating $I_{opioid}$ and $syn_{opioid}$ independently or in combination (*Figure 7D*; *Figure 7—figure supplement 2*). Each combination of $I_{opioid}$ and $syn_{opioid}$ was repeated over n=8 unique synaptic connectivity patterns. $I_{opioid}$ was varied from 0 pA to 6 pA, which was more than sufficient to strongly suppress the IBI spiking of intrinsically tonic *Oprm1+* model neurons (*Figure 7—figure supplement 2*), consistent with the effects of 300 nM DAMGO in vitro (see *Figure 2*). $syn_{opioid}$ was varied from 0 to 0.6 (i.e., synaptic strength = 100–40% of baseline), since this value range was sufficient to account for on our in vitro data demonstrating that 300 nM DAMGO suppresses the amplitude of EPSPs driven by *Oprm1+* neurons by ~50% (see *Figure 5D,E*). When varying $I_{opioid}$ and $syn_{opioid}$ independently, we found that neither parameter reliably reproduced the effects of MOR activation in the preBötC. Indeed, as $I_{opioid}$ was increased, burst frequency was only moderately reduced. As synaptic efficacy was reduced, burst amplitude decreased with inconsistent effects on burst frequency and irregularity. In contrast, manipulation of $I_{opioid}$ and $syn_{opioid}$ in combination had consequences for the model network rhythm that were surprisingly like the effects of MOR activation in the preBötC (*Figure 7C,D*). Frequency was reduced, IBI irregularity was increased, and periodic burst failures became apparent (*Figure 7C*). The model network responded similarly to simulated MOR activation with synaptic inhibition blocked (*Figure 7C, D*), consistent with experimental observations suggesting that inhibitory mechanisms do not play a significant role in producing OIRD (*Bachmutsky et al., 2020*; *Gray et al., 1999*). These network-level effects of simulated MOR activation were associated with changes in the spiking activity of model neurons that were also consistent with our experimental results. Specifically, changes in spiking activity occurred primarily during the inter-burst interval, due to a preferential suppression of spiking among *Oprm1+* vs. *Oprm1−* intrinsically tonic neurons (*Figure 7E,F*), whereas spiking during inspiratory bursts was reduced similarly among *Oprm1+* and *Oprm1−* model neurons (*Figure 7G*). These results in silico support important, yet interdependent, roles for both intrinsic and synaptic mechanisms underlying MOR-mediated suppression of rhythmogenesis in the preBötC.

## Discussion

OIRD is a life-threating consequence of clinical and illicit opioid use that stems from the high sensitivity of the respiratory control network to MOR activation. Developing a detailed mechanistic understanding of how opioids disturb rhythmogenesis at the cellular and network level will help facilitate

the development of new strategies to protect against and reverse OIRD. In this study, we show that MOR activation among a subset of *Oprm1+* neurons reduces the number of spikes, while also impairing the ability of each spike to drive synaptic transmission, thereby functionally dis-integrating *Oprm1+* neurons from the recurrently connected preBötC network. We propose that these dual consequences of MOR activation in the preBötC undermine the generally robust nature of the respiratory rhythm, making it particularly vulnerable to exogenous opioids.

Breathing must be integrated with complex volitional and reflexive behaviors including vocalization, feeding, sensory exploration, and exercise. As such, the inspiratory-rhythm-generating network must be assembled in a way that allows it to be extremely flexible yet reliable in order to reconcile these complex behaviors with the inexorable physiological requirement for gas exchange. At the core of this network, is the preBötC (*Baertsch et al., 2019*; *Smith et al., 1991*). The preBötC contains an intermingled population of neurons with heterogeneous gene expression and connectomes that interact to produce distinct firing phenotypes or 'discharge identities' (*Segers et al., 2012*; *Lalley and Mifflin, 2017*; *Baertsch et al., 2018*). Collectively, this spiking activity results in a rhythmogenic process that can be divided into three functionally distinct phases: a refractory phase, a percolation phase, and a burst phase. Each phase of the inspiratory rhythm can be differentially regulated by neuromodulators (*Baertsch and Ramirez, 2019*) or synaptic inputs (*Zuperku et al., 2019*) to drive dynamic changes in the frequency and regularity of inspiration. For example, the excitatory neuromodulator and potent respiratory stimulant, substance P (SP), specifically affects the percolation phase of the rhythm by increasing the spiking activity of pre-inspiratory neurons between, but not during, bursts. This phase-specific change in spiking activity reduces the time and variability of the IBI, leading to a faster and more regular inspiratory rhythm (*Baertsch and Ramirez, 2019*).

Here we explored this concept in the context of OIRD. Based on single-unit recordings, we find that ~50% of functionally identified preBötC neurons express the *Oprm1* gene, consistent with expression patterns of *Oprm1* determined by ISH (Allen Brain Atlas; *Figure 1—figure supplement 1*) and transgenic mice with a fluorescent reporter fused to the MOR (*Erbs et al., 2015*). Our single-unit recordings also indicate that *Oprm1* expression is evenly distributed among preBötC neuron discharge identities, including putatively rhythmogenic pre-inspiratory neurons (*Ashhad and Feldman, 2020*; *Del Negro et al., 2010*; *Kam et al., 2013a*). Indeed, *Oprm1*-expressing neurons are critical for preBötC rhythmogenesis since pharmacological MOR activation in this isolated network can cause cessation of the rhythm (*Bachmutsky et al., 2020*; *Gray et al., 1999*; *Montandon et al., 2011*; *Wei and Ramirez, 2019*; *Mellen et al., 2003*). Perhaps surprisingly, our data indicate that opioid concentrations that are sufficient to induce severe OIRD did not silence *Oprm1+* preBötC neurons but reduced spiking activity of pre-inspiratory neurons preferentially during the period between inspirations. These opioid-induced changes in spiking activity were related to each individual neuron's intrinsic activity when deprived of synaptic inputs (*Figure 8*). Interestingly, most pre-inspiratory neurons intrinsically produced tonic spiking activity in the absence of excitatory synaptic input. This is notable because it suggests that for these neurons the pre-inspiratory ramp pattern is primarily driven by recovery from intrinsic refractory properties (*Baertsch et al., 2018*; *Krey et al., 2010*) rather than mechanisms of recurrent synaptic excitation. Consistent with the pre-inspiratory spiking activity of these neurons being driven by intrinsic mechanisms, we found that MOR activation had a much greater effect on pre-inspiratory spiking of *Oprm1+* vs. *Oprm1−* intrinsically tonic neurons (*Figure 2D* and *Figure 8*). In contrast, among intrinsically quiescent neurons where excitatory synaptic input is the primary driver of pre-inspiratory spiking, MOR activation produced a substantial suppression of pre-inspiratory spiking regardless of *Oprm1* expression (*Figure 2F* and *Figure 8*). Thus, OIRD involves reduced spiking during the percolation phase due, in part, to suppression of intrinsically driven pre-inspiratory spiking of *Oprm1+* neurons and de-recruitment of neurons with synaptically driven, pre-inspiratory spiking activity. Thus, we predict that opioid-induced suppression of preBötC activity during the percolation phase underlies the prolonged and irregular durations between inspiratory efforts that are characteristic of OIRD in both mice and humans (*Walker et al., 2007*; *Wu et al., 2020*; *Varga et al., 2020*).

These effects on preBötC spiking activity are largely opposite to those produced by the respiratory stimulant SP (*Baertsch and Ramirez, 2019*), and, interestingly, MOR activation can inhibit the release of SP in the spinal cord (*Chen et al., 2018*). Yet, SP only partially reverses OIRD in brainstem slices (*Sun et al., 2019*). The preferential suppression of intrinsically driven IBI spiking we observed among *Oprm1+* pre-inspiratory neurons (*Figure 2*) is consistent with opioids causing

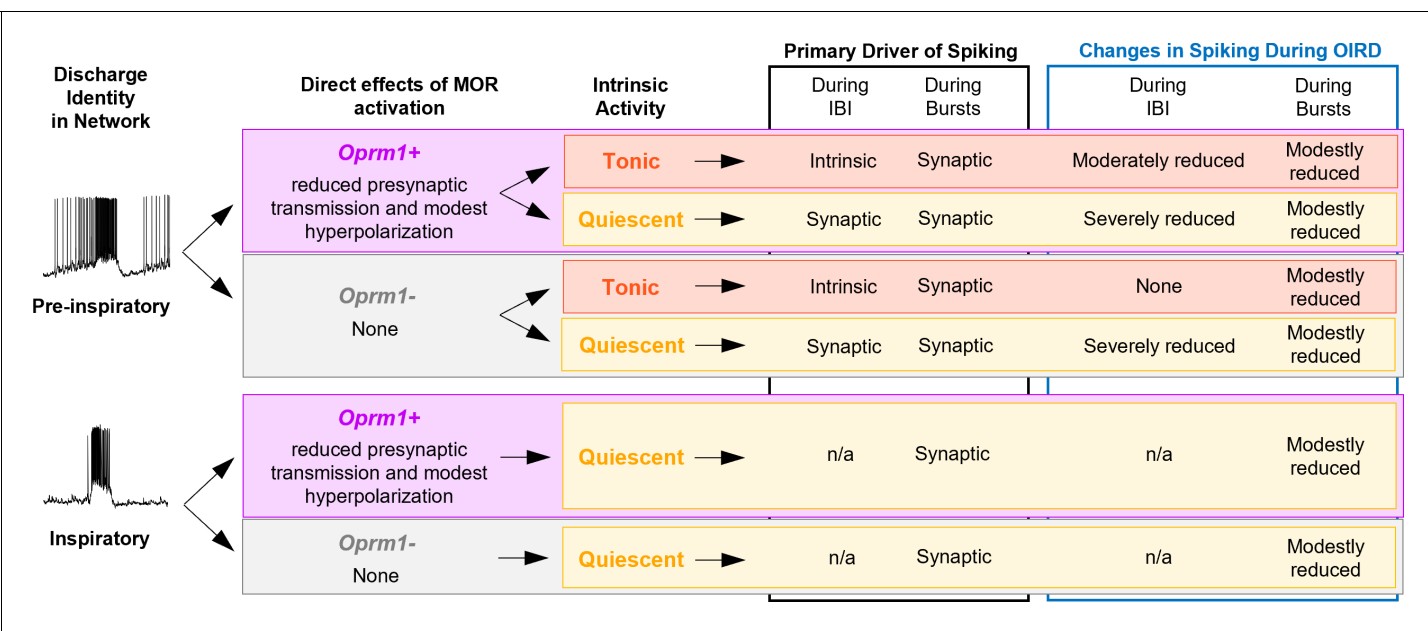

**Figure 8.** Summary of the effects of MOR activation on different types of pre-inspiratory and inspiratory preBötC neurons.

hyperpolarization of MOR-expressing neurons (*Gray et al., 1999*; *Montandon et al., 2011*). However, mimicking changes in preBötC spiking activity during OIRD via hyperpolarization of *Oprm1+* preBötC neurons causes only a moderate suppression of the inspiratory rhythm in vitro and in silico and is even less consequential in vivo (*Figure 4*), suggesting that additional mechanisms play an important role (*Ren et al., 2019*; *Wei and Ramirez, 2019*; *Akins and McCleskey, 1993*).

Indeed, we find the effects of exogenous opioids in the preBötC go beyond changes in spiking activity. MOR activation also weakened excitatory synaptic interactions in the preBötC (*Figure 5D, E*), similar to previous observations in spinal nociceptive pre-synaptic terminals (*Corder et al., 2017*), as well as neurons in the PBN (*Cramer et al., 2021*) prefrontal cortex (*Yamada et al., 2021*), periaqueductal grey (*Lau et al., 2020*), and hippocampus (*Lu et al., 2021*). As a result, the remaining spiking activity of *Oprm1+* preBötC neurons becomes less consequential for network function (*Figure 6*). Thus, considering the recurrent connectivity within the preBötC (*Guerrier et al., 2015*; *Rubin et al., 2009*; *Yang and Feldman, 2018*; *Yang et al., 2020*), it is not surprising that the suppressive effect of opioids on synaptically driven pre-inspiratory spiking is not specific to MOR-expressing preBötC neurons (*Figures 2E* and *5E*). Importantly, the efficacy of spiking activity could not be restored by depolarizing *Oprm1+* preBötC neurons, suggesting the pre-synaptic effect of MOR activation in the preBötC is independent from changes in membrane potential. Modeling these consequences of MOR activation using in silico networks based on the robust and widely utilized Butera-type models (*Butera et al., 1999a*; *Butera et al., 1999b*; *Park and Rubin, 2013*), revealed striking similarities with our experimental findings. In model networks, an *Oprm1+* subpopulation of neurons implemented based on our in vitro data (see *Figure 1*) was necessary for rhythmogenesis (*Figure 7—figure supplement 1*). However, independent manipulations to mimic the hyperpolarization induced changes in spiking activity vs. suppression of pre-synaptic output could not reliably reproduce the effects of MOR activation in the preBötC. Yet, when implemented in combination, the rhythm produced by in silico networks was substantially slowed, with long, irregular periods between bursts. Model networks also exhibited failed bursts reminiscent of our findings in vitro (see *Figure 3D*), which may represent the emergence of mixed-mode oscillations as previously shown in heterogeneous networks with sparse synaptic connectivity (*Bacak et al., 2016*). Exploring the cellular-level consequences of MOR activation using other in silico networks that produce rhythm using distinct biophysical mechanisms (e.g., *Guerrier et al., 2015*) may reveal additional insights into the etiology of OIRD.

The consequences of MOR activation on pre-synaptic function may help explain mechanistically the effectiveness of ampakines for reversing OIRD (*Ren et al., 2009*; *Ren et al., 2015*; *Lorier et al.,*

*2010*; *Greer and Ren, 2009*) because potentiation of post-synaptic AMPAR currents could compensate for opioid-induced impairments in excitatory pre-synaptic transmission. However, ampakine treatment would not be expected to reverse the intrinsic hyperpolarizing effect of opioids on *Oprm1+* preBötC neurons, which could underlie the limitations for ampakine-mediated prevention of OIRD seen clinically (*van der Schier et al., 2014*; *Oertel et al., 2010*). Thus, combination therapies that help the inspiratory rhythm-generating network compensate for both the intrinsic and synaptic consequences of MOR activation in the preBötC may prove to be a more effective strategy for the prevention and reversal of OIRD without affecting analgesia.

MOR activation also caused a moderate reduction in spiking activity during inspiratory bursts. Because preBötC neurons receive strong synchronized excitatory synaptic input during inspiratory bursts (*Ashhad and Feldman, 2020*), we suspect that the reduced spike frequency during bursts is primarily due to the suppressive effect of opioids on pre-synaptic transmission (*Figure 5D,E*). Indeed, spiking of *Oprm1+* and *Opmr1−* neurons during inspiration was similarly reduced by DAMGO (*Figure 2G,H*), suggesting that direct effects of MOR activation on the intrinsic excitability of *Oprm1+* neurons plays a minimal role. Suppressed spiking of single preBötC neurons during inspiratory bursts due to impaired synaptic communication may also contribute to changes in the rate of burst onset observed at the network-level following MOR activation in vivo (*Figure 3J*; *Ferguson and Drummond, 2006*) and/or the higher proportion of failed bursts observed in vitro (*Figure 3D,E*). As observed following chronic exposure to intermittent hypoxia (CIH), weakened preBötC activity during the burst phase can lead to periodic and 'all or none' type failures of inspiratory drive to be transmitted to the XII motor pool (*Garcia et al., 2019*; *Garcia et al., 2017*; *Garcia et al., 2016*). Indeed, MOR activation reduces inspiratory synaptic drive to XII motor neurons without changing their membrane input resistance, and this effect is reversed by ampakines (*Lorier et al., 2010*), consistent with impaired pre-synaptic transmission in *Oprm1+* preBötC neurons (*Figure 5D, E*). Because XII output is important for maintaining upper airway patency, this may have important implications in the context of sleep apnea, which is highly prevalent among opioid users (*Mogri et al., 2009*; *Rose et al., 2014*; *Walker et al., 2007*). Thus, we speculate that the risk of OIRD is amplified by a dangerous feed-forward process that arises due to the synergy between opioids, sleep apnea, and CIH (*Lam et al., 2016*; *Overdyk et al., 2014*).

Our collective results indicate that only ~50% of preBötC neurons express the *Oprm1* gene and are therefore able to play a direct role in OIRD. Yet, respiratory rhythmogenesis is effectively inhibited because opioids act with a 'double-edged sword' to functionally remove *Oprm1* neurons from the preBötC network. By simultaneously reducing intrinsic excitability and impairing excitatory synaptic communication, exogenous opioids disable the normally robust inspiratory network, leading to long and variable pauses between breaths and ultimately cessation of rhythmogenesis altogether. Thus, interesting, yet unresolved, questions are as follows: Why do these critical rhythm-generating neurons express the MOR? Are there circumstances when the heightened opioid sensitivity of the preBötC may provide and evolutionary advantage? And how and when might endogenous opioid signaling play important functional roles in the preBötC? Addressing these important questions may reveal naturally occurring mechanisms or neuromodulatory states that protect this vital respiratory network from opioids. Such insights will be a critical next step in the development of novel strategies to protect against and reverse OIRD.

## Materials and methods

### Animals

Experiments were performed on neonatal (p4–p12) and adult (>p40) male and female C57Bl/6 mice bred at Seattle Children's Research Institute. Homozygous *Oprm1^CreGFP* mice were provided by the laboratory of Dr. Richard Palmiter (University of Washington) (Jax Stock No: 035574). *Oprm1^CreGFP* were genotyped using a standard protocol with the following three primers: 5' CCT TCC ACT CAG AGA GTG GCG (*Oprm1* forward), 5' CCT TCC ACT CAG AGA GTG GCG (*Oprm1* reverse), and 5' GGC AAA TTT TGG TGT ACG GTC AG (*Cre* reverse). The wild-type allele gives a band of ~500 bp, while the targeted allele gives a band of ~400 bp after 34 cycles with 20 s annealing at 60°C. *Oprm1-^CreGFP* mice were crossed with homozygous mice derived at the Allen Brain Institute containing either (1) floxed STOP channelrhodopsin2 fused to EYFP, *Rosa26^Isl-ChR2:EYFP*, or more simply

$Rosa26^{ChR2}$ (Ai32, Jax Stock No: 024109) or (2) floxed STOP ArcherhodopsinT fused to EYFP, $Rosa26^{lsl-ArchT:EYFP}$, or more simply $Rosa26^{ArchT}$ (Ai40D, Jax Stock 021188). All mice were group housed with access to food and water ad libitum in a temperature-controlled (22 ± 1°C) facility with a 12 hr light/dark cycle.

## In vitro medullary horizontal slice preparation

Horizontal medullary slices containing the ventral respiratory column were prepared from postnatal day 4–12 mice as previously described (*Baertsch et al., 2019*). Briefly, whole brainstems were dissected in ice-cold, artificial cerebrospinal fluid (aCSF; in mM: 118 NaCl, 3.0 KCl, 25 NaHCO₃, 1 NaH₂PO₄, 1.0 MgCl₂, 1.5 CaCl₂, 30 D-glucose) equilibrated with carbogen (95% O₂, 5% CO₂). aCSF had an osmolarity of 305–312 mOSM and a pH of 7.40–7.45 when equilibrated with gas mixtures containing 5% CO₂ at ambient pressure. Cyanoacrylate was used to secure the dorsal surface of the brainstem to an agar block cut at a ~15° angle, and a vibratome was used (Leica 1000S) to section the brainstem in the transverse plane in 200 µm steps moving from rostral to caudal until the VII nerves were visualized. Brainstems were then sectioned in the horizontal plane by reorienting the agar block to position its ventral surface facing up. The blade was leveled with the ventral edge of the brainstem, and a single ~850 µm step was taken. The angle of the horizontal section through the tissue is determined by the angle at which the agar block was cut and is critical for cutting a horizontal slice at the correct thickness. A partially open 'teardrop'-shaped central canal is indicative of a properly prepared horizontal slice. The preBötC is located lateral to the rostral end of the teardrop and approximately ½–¾ of the distance between the midline and the lateral edge of the tissue.

Slices were placed in a custom recording chamber containing circulating aCSF (~15 ml/min) warmed to 30°C. The [K+] in the aCSF was then gradually raised from 3 to 8 mM over ~10 min to elevate neuronal excitability. Glass pipette electrodes (<1 MΩ tip resistance) filled with aCSF were placed on the surface of the slice to record rhythmic extracellular neuronal population activity. Signals were amplified 10,000×, filtered (low pass, 300 Hz; high pass, 5 kHz), rectified, integrated, and digitized (Digidata 1550A, Axon Instruments). The blind patch clamp approach was used to record the activity of single neurons. Recording electrodes were pulled from borosilicate glass (4–8 MΩ tip resistance) using a P-97 Flaming/Brown micropipette puller (Sutter Instrument Co., Novato, CA) and filled with intracellular patch electrode solution containing (in mM): 140 potassium gluconate, 1 CaCl₂, 10 EGTA, 2 MgCl₂, 4 Na₂ATP, and 10 Hepes (pH 7.2). To map the location of recorded neurons, patch pipettes were backfilled with intracellular patch solution containing 2 mg/ml Alexa Fluor568 Hyrdazide (ThermoFisher). Neuronal activity was recorded in current clamp mode in either cell-attached or whole-cell configuration (depending on the specific experiment as noted in the text) using a MultiClamp 700B amplifier (Molecular Devices, Sunnyvale, CA). Extracellular population activity and intracellular signals were acquired with pCLAMP software (Molecular Devices, Sunnyvale, CA). After cell-attached recordings, the neuronal membrane was ruptured to allow the AlexaFluor fluorescent maker to fill the cell body. Following each experiment, the dorsal surfaces of fresh, unfixed slices were imaged (2.5×) using a Leica DM 4000 B epifluorescence microscope equipped with 405, 488, and 548 nm laser lines. Images were post-processed using Image-J software (Version 1.48); brightfield and epifluorescent images of Alexa Fluor 568 labeled cell(s) were overlayed to determine the coordinates of the recorded neuron(s) relative to rostral edge of the slice (VII nerve; Y direction) and the midline (X direction) (*Figure 1D,E*).

## In vivo surgical preparation

Adult mice were induced with isoflurane (~3%) and then transferred to urethane anesthesia (1.5 g/kg, i.p.). Mice were then placed supine on a custom heated surgical table to maintain body temp at ~37°C. The trachea was exposed through a midline incision and cannulated with a curved (180 degree) tracheal tube (24 G) caudal to the larynx and then mice spontaneously breathed 100% O₂ throughout the remainder of the surgery and experimental protocol. ECG leads were placed on the fore and hind paw to monitor heart rate. The trachea and esophagus were removed rostral to the tracheal tube, and the underlying muscles were removed to expose the basal surface of the occipital bone. The portion of the occipital bone and dura overlying the ventral medullary surface were removed, and the exposed surface of the brainstem was superfused with warmed (~37°C) aCSF equilibrated with carbogen (95% O₂, 5% CO₂). The hypoglossal nerve (XII) was isolated unilaterally,

cut distally, and recorded from using a suction electrode connected to a fire-polished pulled glass pipette filled with aCSF. To record multi-unit neuronal population activity directly from the preBötC, tapered pulled glass pipettes with a sharp broken tip (<1MΩ tip resistance) filled with aCSF were advanced into the ventral medulla ~200–500 µm until integrated rhythmic activity was maximal. Electrical activity from the brainstem and XII nerve was amplified (10,000X), filtered (low pass 300 Hz, high pass 5 kHz), rectified, integrated, and digitized (Digidata 1550A, Axon Instruments). In some experiments, extracellular activity was recorded from single units in vivo using 4–8 MΩ pulled glass electrodes filled with aCSF. Prior to experimental protocols, adequate depth of anesthesia was determined via heart rate and respiratory responses to toe pinch and adjusted if necessary with supplemental urethane (i.p.).

## Optogenetic and pharmacological manipulations

Two hundred micrometer diameter glass fiber optics (0.24 NA) connected to blue (470 nm) high-powered LEDs or yellow-orange (598 nm) lasers were positioned above the preBötC either bilalaterally or ipsilateral/contralateral to the population and/or unit recordings depending on the specific experiment (as indicated in the text and figure legends). Light power was calibrated using an optical power meter (ThorLabs). Powers and durations of light pulses are noted in the text and figures. During single-unit recordings in vitro using the blind patch approach, neurons were classified as $Oprm1$ + based on optogenetic responses. In $Oprm1^{CreGFP}$; $Rosa26^{ArchT:EYFP}$ slices, neurons that were inhibited during 598 nm light were designated as $Oprm1+$, whereas those that lacked a response were presumed to be $Oprm1-$. In $Oprm1^{Cre:GFP}$; $Rosa26^{ChR2:EYFP}$ slices, Oprm1 expression was determined using one or both of the following methods: (1) the presence of an excitatory response to light following pharmacological blockade of glutamatergic synaptic transmission (20 µM CNQX, 20 µM CPP) and (2) the presence of spikes generated reliably and with short latency ~5–10 ms following brief 10 ms 470 nm light pulses ipsilateral to the recording electrode (e.g., see *Figure 5A–C*). In many cases, these strategies were used in combination to characterize neurons as $Oprm1+$ or $Oprm1-$.

During in vitro experiments, stable baseline preBötC population and single-unit activities were recorded for ≥5 min prior to addition of the MOR-agonist DAMGO ([D-Ala2, N-Me-Phe4, Gly5-ol]-Enkephalin) to the circulating aCSF bath. DAMGO (Sigma Aldrich) stock solutions (1 mM in $H_2O$) were aliquoted, and stored at −20°C. In dose-response experiments, DAMGO was added to the aCSF bath at total concentrations of 50, 100, 200, and 300 nM at 7 min intervals, and data were analyzed during the last 2 min of each interval. Each slice preparation only received a single step-wise exposure to DAMGO. In some experiments, MOR activation was reversed with the competitive MOR antagonist Naloxone (Nx) and AMPAR- and NMDAR-dependent glutamatergic synaptic transmission was blocked by adding CNQX (6-cyano-7-nitroquinoxaline-2,3-dione disodium) and (R)-CPP (3-((R)−2-carboxypiperazin-4-yl)-propyl-1-phosphonic acid) to the aCSF bath. Naloxone (Tocris) was diluted to a 100 mM stock solution in $H_2O$. CNQX and CPP (Tocris) were diluted to 20 mM stock solutions in $H_2O$. All drugs were aliquoted and stored at −20°C. For experiments in vivo, stable baseline preBötC and XII multi-unit activity was established for ≥5 min prior to systemic injection (i.p.) of morphine (150 mg/kg; Patterson Veterinary Supply).

## Constructing a computational model of OIRD in the preBötC

The computational model of the preBötC consisted of 300 Hodgkin–Huxley style neurons with equations modified from *Butera et al., 1999a*; *Butera et al., 1999b*. The membrane voltage of each neuron is given by:

$$-C_m \frac{dv}{dt} = I_{Na} + I_K + I_{leak} + I_{NaP} + I_{opioid} + I_{syn}$$

where the currents are

$$I_{Na} = g_{Na} \cdot m_\infty^3 \cdot (1-n) \cdot (v - E_{Na})$$

$$I_K = g_K \cdot n^4 \cdot (v - E_K)$$

$$I_{NaP} = g_{NaP} \cdot m_{NaP_\infty} \cdot h \cdot (v - E_{Na})$$

$$I_{leak} = g_{leak} \cdot (v - E_{leak})$$

$$I_{opioid} = \begin{cases} I_{opioid}, & Oprm1^+ \\ 0pA, & Oprm1^- \end{cases}$$

And:

$$\frac{dn}{dt} = \frac{(n_\infty - n)}{\tau_n}$$

$$\frac{dh}{dt} = \frac{(h_\infty - h)}{\tau_h}$$

$$m_\infty = \frac{1}{1 + e^{\left(\frac{v - v_m}{\sigma_m}\right)}}$$

$$n_\infty = \frac{1}{1 + e^{\left(\frac{v - v_n}{\sigma_n}\right)}}$$

$$m_{NaP_\infty} = \frac{1}{1 + e^{\left(\frac{v - v_{m_{NaP}}}{\sigma_{m_{NaP}}}\right)}}$$

$$h_\infty = \frac{1}{1 + e^{\left(\frac{v - v_h}{\sigma_h}\right)}}$$

$$\tau_n = \frac{\tau_{nb}}{cosh\left(\frac{v - v_n}{2\sigma_n}\right)}$$

$$\tau_h = \frac{\tau_{hb}}{cosh\left(\frac{v - v_h}{2\sigma_h}\right)}$$

The synaptic currents for neuron post-synaptic neuron $i$ are given by:

$$I_{syn,i} = \sum_{j \in OPRM1^+:j \to i} (1 - syn_{opioid}) \cdot g_E s_{ij} (V_i - E_{synE})$$
$$+ \sum_{j \in OPRM1^-:j \to i} g_E s_{ij} (V_i - E_{synE}) + \sum_{j \in I:j \to i} g_I s_{ij} (V_i - E_{synI})$$

So that if pre-synaptic neuron $j$ is $Oprm1^+$, the excitatory conductance of the $j \to i$ synapse is scaled by the controlled parameter $syn_{opioid} = [0,1]$. The dynamics of the synapses are governed by:

$$\frac{ds}{dt} = \frac{(1 - s_{ij})m_\infty^{(ij)}(V_j) - s_{ij}}{\tau_{syn}}$$

$$m_\infty^{(ij)}(V_j) = \frac{1}{1 + e^{\frac{V_j - \theta_{syn}}{\sigma_{syn}}}}$$

Cellular parameters are listed in *Supplementary file 1*, and network parameters are listed in *Supplementary file 2*.

We vary $I_{opioid}$ from 0 to 6 pA in steps of 0.5 pA and $syn_{opioid}$ from 0 to 0.6 in steps of 0.05. $I_{opioid}$ and $syn_{opioid}$ were set to zero during 'control' conditions and to a proscribed value during simulated MOR activation. Connectivity was generated randomly with probability $p = \frac{k_{avg}}{2(N-1)}$ where

$k_{avg} = 6; N = 300$. Eight replicates were performed of each combination of $I_{opioid}$ and $syn_{opioid}$ by initializing the random number generator with an integer seed in [0,7]. All simulations were performed in python using Brian2, and code is available at https://github.com/nbush257/oprm1 (*Baertsch, 2021* copy archived at swh:1:rev:feeed0404ade9d4155ea9f6e29e0f4ec1faf57f7) and upon request.

Simulated population rates were smoothed with a Gaussian kernel with $\sigma = 25ms$, and bursts were defined as excursions of the population rate above 10 sp/s.

## Quantification and statistical analysis

Effects of MOR activation on the spiking activity of individual preBötC neurons (*Figure 2*) were quantified from ~10 to 20 consecutive respiratory cycles during the last 2 min of each dose of DAMGO. Action potentials and integrated preBötC population bursts were detected using the threshold search function in Clampfit (Molecular Devices). Spike times were compared to onset and offset times of preBötC population bursts to quantify spike frequencies specifically between (inter-burst interval) or during bursts. For pre-inspiratory neurons, average spiking frequency at baseline and in 300 nM DAMGO was also quantified over the course of the inter-burst interval. Spike times were normalized relative to the duration of each IBI, and instantaneous spike frequencies were averaged within 1000-time bins between the start and end of the IBI. To quantify changes in total preBötC population spiking in vitro and in vivo between inspiratory bursts (IBI) in response to MOR activation, background noise was measured within the aCSF bath, but prior to contacting the surface of slices with the recording electrode. This noise value was then subtracted from the amplitude of the integrated population recording during ~10–20 consecutive IBIs to estimate 'total spiking activity' during this period (*Figure 3A*). IBI spiking was then measured during subsequent MOR activation or optogenetic manipulations and normalized to the baseline value. Failed bursts were distinguished from successful bursts as being >2 standard deviations from the mean burst amplitude. MOR-induced changes in evoked EPSP amplitudes (*Figure 5D,E*) were determined by unilaterally stimulating *Oprm1+* preBötC neurons while recording $V_m$ from neurons in the contralateral preBötC. In responsive neurons, 50–100 trials containing a 10 ms light pulse were delivered at baseline, and in 50 and 300 nM, DAMGO and evoked EPSP amplitudes were averaged across all trials. Irregularity in frequency and amplitude was calculated as follows: $irreg\ score = ABS\left(\frac{N-(N-1)}{N}\right)$.

Statistical comparisons were performed using GraphPad Prism8 software. Groups were compared using appropriate two-tailed t-tests or one-way or two-way ANOVAs with Bonferroni's multiple comparisons post hoc tests. Differences were considered significant at p<0.05, and data are displayed as means ± standard error. For all post hoc statistical comparisons, p-values are designated in the figures as follows: *p<0.05, #p<0.01, †p<0.001, ‡p<0.0001. Data was visualized, and figures were assembled using a combination of Clampfit, GraphPad, and PowerPoint software.

## Acknowledgements

We thank Dr. Richard Palmiter and his lab for the development and generous donation of the *Oprm1*[Cre:GFP] mouse line, Dr. Aguan Wei for helpful edits on the manuscript, and the National Heart, Lung, and Blood Institute for funding this work: K99HL145004 (NAB), R00HL145004 (NAB), R01HL144801 (JMR), and F32HL154558 (NJB).

## Additional information

### Funding

| Funder | Grant reference number | Author |
| --- | --- | --- |
| National Heart, Lung, and Blood Institute | K99HL145004 | Nathan A Baertsch |
| National Heart, Lung, and Blood Institute | R01HL144801 | Jan-Marino Ramirez |
| National Heart, Lung, and Blood Institute | F32HL154558 | Nicholas J Burgraff |
| National Heart, Lung, and | R00HL145004 | Nathan A Baertsch |

Blood Institute

The funders had no role in study design, data collection and interpretation, or the decision to submit the work for publication.

## Author contributions
Nathan A Baertsch, Conceptualization, Resources, Formal analysis, Supervision, Funding acquisition, Validation, Investigation, Visualization, Methodology, Writing - original draft, Project administration, Writing - review and editing; Nicholas E Bush, Software, Formal analysis, Investigation, Visualization, Methodology, Writing - review and editing; Nicholas J Burgraff, Funding acquisition, Validation, Investigation, Writing - review and editing; Jan-Marino Ramirez, Resources, Supervision, Funding acquisition, Writing - review and editing

## Author ORCIDs
Nathan A Baertsch ⓘ https://orcid.org/0000-0003-1589-5575
Jan-Marino Ramirez ⓘ http://orcid.org/0000-0002-5626-3999

## Ethics
Animal experimentation: This study was performed in accordance with the NIH Guide for the Care and Use of Laboratory Animals and approved institutional animal care and use committee (IACUC) protocols at Seattle Children's Research Institute (Protocol ID: IACUC00058).

## Decision letter and Author response
Decision letter https://doi.org/10.7554/eLife.67523.sa1
Author response https://doi.org/10.7554/eLife.67523.sa2

# Additional files

## Supplementary files
- Supplementary file 1. Network parameters for in silico preBötC network.
- Supplementary file 2. Cellular parameters for in silico preBötC neurons.
- Transparent reporting form

## Data availability
All data generated or analyzed during this study are included in the manuscript and supporting files.

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
