## [Decision Letter]

**Acceptance summary:**

Opioids are widely used as pain killers, but present the severe side-effect of respiratory depression. The study from Baertsch et al., provides a mechanistic understanding of the actions of opioids on breathing by elucidating some of the biophysical and synaptic mechanisms by which opioids depress breathing with the goal of identifying therapeutic strategies. The data suggest that opioid-induced respiratory depression (OIRD) is due to both presynaptic hyperpolarization, and reduction of synaptic efficacy. The data presented here definitively advance our understanding of the mechanisms of OIRD at the level of central respiratory neural circuits.

**Decision letter after peer review:**

Thank you for submitting your article "Dual mechanisms of opioid-induced respiratory depression in the inspiratory rhythm generating network" for consideration by *eLife*. Your article has been reviewed by 3 peer reviewers, and the evaluation has been overseen by a Reviewing Editor and Ronald Calabrese as the Senior Editor. The following individuals involved in review of your submission have agreed to reveal their identity: Gaspard Montandon (Reviewer #1); Jeffrey C Smith (Reviewer #3).

Essential Revisions:

1 – Animal model validation: First, the authors must provide (1) anatomical/histological controls of opsin channel expression in Oprm1 neurons in the preBötC region (nb of neurons indeed expressing the opsin among the targeted population (Oprm1+ neurons), level of opsin expression). Was a fluorescent molecule inserted with the opsin channels for the in-vivo experiments to verify expression? Also, control experiments with wild-type animals are usually performed to confirm that it was not the light's heat altering neuronal activity. Was this done? Second, we need some functional controls showing that the expression of the opsin channels is high enough to mimic the known DAMGO effects (for example is the photoinhibition inducing a membrane hyperpolarization comparable to that produced by DAMGO).

2 – Computational modeling results: The reviewers have concerns about the lack of discussion of biophysical mechanisms contributing to preBötC rhythm generation and their lack of justification for the model formulation, including discussing why the model was parameterized as indicated. Even if the model chosen (the Butera's model) and its dynamics are probably not a good representation of preBötC circuit dynamics in vivo, we suppose that the authors got their implementation of that model to show the basic result that a combination of neuronal hyperpolarization and depression of excitatory synaptic transmission could qualitatively mimic the in vitro results. We felt that they need to explain and justify the parameter sets used, and also include in the Discussion why they believe that the model is appropriate. They should also note that the plausibility of the proposed dual mechanisms of opioid action on the preBötC network needs to be checked with other models (e.g., Guerrier et al.,).

3 – Place the results in a more general context regarding known inhibitory effects of opioids in other circuits and discuss the potential differences between mice and human in that regard. Also, discuss the data obtained in young animals in vitro versus data from adult rodents in vivo.

4 – Attention should be put on statistics that are not clear enough in the present state (P values and tests should be given everywhere and must fit the illustrations, the text and the legends). Also Details on the number of animals used are missing. More importantly, an important issue is whether the authors have adequately considered hysteresis effects associated with opioid exposure. This is critical, because the findings of this study have implications for a serious public health issue. For instance, if the dose-response component of their study included randomization of opioid concentration presentation order, then the Authors can analyze their data to assess whether response to 300 nM DAMGO in a naïve animal matches responses to the same concentration in an animal previously exposed to lower dosages. In contrast, if dosages were always presented in the same order, then it would be informative if they carried out experiments in which 300 nM DAMGO was applied to naïve animals, and compared responses to responses they've already obtained in their dose-response studies. If responses are not significantly different, then this issue can be disregarded. This should be at least discussed.

*Reviewer #1 (Recommendations for the authors):*

Baertsch et al. present a very compelling study investigating network level mechanisms underlying opioid-induced respiratory depression. This study is topical considering the respiratory side-effects of opioids and their substantial misuse in North America. The authors propose that MORs inhibit a subset of neurons expressing Oprm1 and that respiratory network inhibition is due to reduction of pre-inspiratory spiking and inhibition of excitatory synaptic transmission. The study is mechanistic, it is new and, while the experiments are technically challenging, they are overall well-executed. There are limitations to the study that preclude its publication in its current state. The major limitations are related to data analysis, presentation, and interpretation. The comments below aim to improve the study and can be addressed by the authors. Some controls such as histological analysis of opsin channel expression in Oprm1 neurons in the region of the preBötC may be necessary. Overall, there all a lot of data presented and it may be difficult for some readers to follow the discussion. The authors could simplify the results and discussion to make easier to understand for a general audience. However, I believe that all the comments listed below can be addressed by the authors.

1) The main conclusion of Figure 4 is that inhibition of Oprm1 cells in vitro and in vivo do not replicate the effects of DAMGO or morphine. Or that hyperpolarization of oprm1 cannot fully account for OIRD. Although I tend to agree with this conclusion, how can the authors be sure that all Oprm1+ cells expressed opsin channels, or that a sufficient amount of channels is expressed? One challenge with cre-lox recombination approaches is to demonstrate whether the cells expressing cre recombinase are expressing ArchT. If the Oprm1/cre expression is relatively low, then the opsin channels may not be highly expressed which would limit the inhibition by light. What is the expression of ArchT in Oprm1+ neurons? Is the expression of the opsin channels high enough to match the effects of MOR inhibition by DAMGO. The authors need to demonstrate that all neurons expressing Oprm1 are expressing ArchT at a sufficient level. Was there some staining or microscopy done to confirm this? These are critical control that need to be performed for all experiments using optogenetics.

2) Neonatal in vitro versus adult in vivo. The authors often compare experiments done in neonatal in-vitro preparations with adult in vivo experiments. The authors should mention clearly what are the potential differences between these experiments: immature respiratory network, differences in opioid sensitivity, raised potassium, room temperature versus adult network, anesthesia etc. This should be discussed.

3) Statistical comparisons. There are many experiments where some two-way ANOVAs were shown in figure legends, but the p-values of post-hoc tests are not presented. A lot of percentages are presented without statistical tests. This need to be addressed. For instance Figure 4C. It is said in text that DAMGO has an effect on IBI but it is not shown with a * in Figure. P-values and n should be consistently presented in text or legends. This applies to all figures.

4) I would suggest to the authors to add to Figure 7 (or a separate Figure 8) a schematic of types of preBötC neurons affected by DAMGO. It may be difficult to the readers to conceptualize the impact of DAMGO on preBötC neurons without seeing a summary of different roles and responses of neurons. Could the authors present a table summarizing the impact of DAMGO on pre-inspiratory, inspiratory, oprm1- and oprm1- neurons? Or mechanisms. Or a diagram?

5) Presentation of data and results. The presentation of the data could be improved. Indeed, the important results presented in many figures can be hard to identify. For instance, in figure 2, panel C is very small and not highlighted. If the authors could find a way to increase the size of panels, it would greatly improve their impacts. This applies to all figures.

6) The mechanisms identified here should be compared to existing work related to opioid-mediated inhibition for other model systems such as MOR inhibition in pain circuits.

7) The authors should comment and discuss the fact that opioid drugs such as morphine and fentanyl may have different impacts in mice compared to humans, and that dose-dependent effects are likely observed as previously shown by other authors (Albert Dahan etc.)

*Reviewer #2 (Recommendations for the authors):*

1. Although the number of cells recorded from is documented, the number of animals used is not mentioned anywhere in the manuscript. How many opioid exposures were carried out in each experiment? Was the order of presentation of different DAMGO concentrations varied? What was the interval between DAMGO exposures?

2. 4. The Butera model of respiratory rhythmogenic networks is computationally tractable, high-level, and low-dimensional. Thus, it is appealing because it helps us develop intuitions about this rhythmogenic network. In the original model, pacemaker neurons were obligatory constituents. At the time the model was developed, this was unproblematic, because there was broad consensus that respiratory rhythm arose out of the activity of a kernel of endogenously bursting (EB) pacemaker neurons. As anyone who has followed the literature on this topic knows, the existence and significance of EBs within the PBC has become contentious, with both their relative number and functional significance brought into question.

It is puzzling that the Authors didn't implement their in silico simulation based on a more recent model that does not require the presence of EBs, and maintains stable rhythmicity via synaptic interactions between pre-I neurons (Guerrier et al., 2015). This model is an attractive alternative, because phasic variation in synaptic efficacy is the mechanism for rhythmogenesis. Thus, the dual action of opioids on presynaptic excitability and post-synaptic efficacy proposed here as the mechanism for OIRD could straight-forwardly be implemented.

*Reviewer #3 (Recommendations for the authors):*

This study investigates the important problem of network- and cellular-level mechanisms of opioid-induced respiratory depression (OIRD) operating within the preBötzinger complex inspiratory rhythm generator, which has been shown to be one of the critical brainstem respiratory regions involved. The authors present a detailed electrophysiological analysis in mice of the spiking patterns of preBötC neurons expressing mu-opioid receptors (MOR) and examined how spiking patterns and preBötC population activity is affected by MOR agonists in vitro and in vivo. The studies incorporated optogenetic approaches to identify by photostimulation neurons expressing the MOR encoding gene Oprm, and to manipulate activity of these neurons by photoinhibition. The authors present important data on the electrophysiological phenotypes of Oprm+ neurons. They present novel evidence that MOR activation reduces inspiratory neuron spiking particularly of neurons with pre-inspiratory spiking patterns and depresses excitatory synaptic transmission- dual mechanisms of opioid action that the authors propose can account for the perturbations of inspiratory rhythm generation. Simulations with a preBötC network biophysical model also suggest that these dual mechanisms in combination are required to explain how inspiratory rhythm generation can be disrupted by MOR activation. The paper is well generally well written and the data presented for the most part advances understanding of the mechanisms of OIRD at the level of preBötC circuits.

1) The authors are proposing that MOR activation-induced hyperpolarization of preBötC excitatory neurons alone is not sufficient to explain disruption of inspiratory rhythm generation. This is based largely on the experimental observations that optogenetic hyperpolarization of Oprm+ preBötC neurons does not produce the same level of depression of preBötC neuronal spiking activity and inspiratory burst frequency as DAMGO. The authors do not present data showing that the depolarization produced by photoinhibition is comparable to that produced by DAMGO. The only figure that could potentially illustrate effects of DAMGO on membrane potential of inspiratory neurons from whole-cell recording is Supplemental Figure 4, which is a nice and necessary validation of their transgenic mouse construct, but this figure does not indicate values of baseline membrane potentials before and after DAMGO. The authors should provide more detailed information on the level of membrane hyperpolarization produced by photoinhibition compared to that produced by DAMGO.

2) The authors present interesting modeling results suggesting that hyperpolarizing model neurons incorporating opioid-induced currents vs. reducing their synaptic output has effects on network spiking activity resembling the differential effects on cellular/network activity observed during optogenetic hyperpolarization of Oprm+ neurons vs. MOR activation. There is no discussion of the modeling results in terms of the model biophysical mechanisms and formulation. This model in general can show that hyperpolarization of key rhythmogenic neurons (e.g., via K^+^-dominated leak currents) can produce the perturbations of the rhythm similar to MOR activation, depending on parameterization of the conductances. The authors present an interesting formulation of the Butera et al. model in terms of the range of NaP conductances and the values of excitatory synaptic conductances, both of which affect the rhythmogenic capabilities of this model. Why was the model parameterized as specified? Why was the paper of Bacak et al., (*eLife* 2016;5:e13403. DOI: 10.7554/*eLife*.13403) not referenced in terms of how cellular activity synchronization due to synaptic connection weights influences the population activity amplitude fluctuations and other aspects of how the basic model employed here produces circuit dynamics relevant to this study?

---

## [Author Response]

Essential Revisions:1 – Animal model validation: First, the authors must provide (1) anatomical/histological controls of opsin channel expression in Oprm1 neurons in the preBötC region (nb of neurons indeed expressing the opsin among the targeted population (Oprm1+ neurons), level of opsin expression). Was a fluorescent molecule inserted with the opsin channels for the in-vivo experiments to verify expression? Also, control experiments with wild-type animals are usually performed to confirm that it was not the light's heat altering neuronal activity. Was this done? Second, we need some functional controls showing that the expression of the opsin channels is high enough to mimic the known DAMGO effects (for example is the photoinhibition inducing a membrane hyperpolarization comparable to that produced by DAMGO).

We thank the editor and expert reviewers for their constructive comments that have helped us improve our manuscript!

We now include control experiments to confirm that light stimulations have no effect on control animals (lacking ChR2 expression) at the network level, both in vitro and in vivo (Figure 6—figure supplement 3). This is in addition to the cellular-level observation that light stimulations have no effect on the Vm of neurons not expressing ChR2 (Figure 6—figure supplement 1).

Please see our detailed response to Reviewer 1 and 3 below regarding our approach to mimic the effects of DAMGO. Briefly, instead of trying to match MOR induced hyperpolarization at the level of single neurons, we used a network-level approach and “matched” the change in integrated network spiking activity induced by 300nM DAMGO with the change in integrated spiking activity during photoinhibition of Oprm1 neurons. in vitro, 2mW was sufficient for this (top panel in Figure 4D), and 6mW was sufficient in vivo (top panel in Figure 4I). At these light intensities, photoinhibition reduced integrated network spiking activity by an amount equivalent to that observed in 300nM DAMGO but did not cause a similar frequency depression. Thus, supporting our conclusion that hyperpolarization alone in not sufficient to mimic OIRD in the preBötC.

Please see our detailed response to Reviewer 1 below regarding opsin expression and histological studies. Briefly, we used a functional electrophysiological strategy to confirm that light sensitive (Cre-expressing) neurons were also sensitive to DAMGO, whereas neurons that were not light sensitive (not Cre-expressing) were much less sensitive to DAMGO (Figure 2D), providing strong evidence that opsin and MOR were expressed on the same preBötC neurons. Further, the Oprm1Cre mouse line has been recently validated by the donating lab (Liu et al., 2021, PNAS). Although we initially considered the suggested immunohistochemistry experiments; this approach has many of its own limitations e.g. antibody specificity, cross-reactivity, background staining, etc., and is not necessarily more specific than the highly efficient intersectional genetic approach used here. Therefore, because of its prior validation and the specificity indicated by our electrophysiological results, we did not perform immunohistochemistry to look for colocalization of MOR immunoreactivity and Cre-reporter expression.

2 – Computational modeling results: The reviewers have concerns about the lack of discussion of biophysical mechanisms contributing to preBötC rhythm generation and their lack of justification for the model formulation, including discussing why the model was parameterized as indicated. Even if the model chosen (the Butera's model) and its dynamics are probably not a good representation of preBötC circuit dynamics in vivo, we suppose that the authors got their implementation of that model to show the basic result that a combination of neuronal hyperpolarization and depression of excitatory synaptic transmission could qualitatively mimic the in vitro results. We felt that they need to explain and justify the parameter sets used, and also include in the Discussion why they believe that the model is appropriate. They should also note that the plausibility of the proposed dual mechanisms of opioid action on the preBötC network needs to be checked with other models (e.g., Guerrier et al.).

The text has been modified at e.g. lines ~415-425 to justify the use of the Butera model and the parameters chosen, and at lines ~565-570 in the Discussion to suggest that this result ought to be tested in other models that utilize distinct biophysical mechanisms.

Indeed, other models can capture detailed dynamics and behaviors of single neurons and the preBötC network in more detail. As the reviewers suppose, the Butera model is chosen as it is simple and effective in producing regular bursting behaviors at the network level, thus allowing for examination of the network level effects of implementing either, or both, neuronal hyperpolarization and synaptic depression. The parameters were modified from previously published parameters that incorporate inhibitory populations (Harris et al., 2017). Inhibitory populations are a key component of the respiratory network, but not included in many published models (including the Guerrier model). Parameters were modified from Harris et al., but NaP and leak conductances were modified to better qualitatively match in-vitro observations of burst frequency and duration. Further, the model was created so that there were no intrinsically bursting cells (i.e., cells do not exhibit bursting behavior in the absence of synaptic connections) We did this because (1) the “critical” significance of endogenous bursters is controversial, (2) in this study, all neurons we found in our in vitro patch experiments were either intrinsically tonic or quiescent (not bursting) (3) the model generated regular bursting without an endogenously bursting subpopulation included, and adding ~5-10% endogenous bursting neurons in the model didn’t significantly impact our results or conclusions.

In contrast to the specific role of endogenous bursting neurons, the presence of INaP within preBötC neurons is well-supported and generally accepted. The Guerrier et al., 2015 model is important because it supports the hypothesis that intrinsic burst-promoting currents (e.g. INaP) may not be required for rhythmogenesis under all conditions, and that synaptic mechanisms per se may be sufficient given certain dynamics. However, this does not mean that INaP doesn’t exist or play an important functional role within the preBötC. Thus, excluding INaP from our computational experiments did not seem appropriate to address our research question, and instead we used an adapted version of the tractable and widely implemented Butera model in order to examine the potential interaction of hyperpolarization and/or presynaptic suppression on the network-level activity.

3 – Place the results in a more general context regarding known inhibitory effects of opioids in other circuits and discuss the potential differences between mice and human in that regard. Also, discuss the data obtained in young animals in vitro versus data from adult rodents in vivo.

We have revised the text to include more background and references related to the effects of opioids in other circuits. e.g. Line ~322:

“The mechanisms of opioid action are diverse and vary based on brain region (Crain and Shen, 1990, Yaksh, 1997, Bourgoin et al., 1994, Christie, 1991). However, in many neuronal circuits including the parabrachial nucleus (Cramer et al., 2021) prefrontal cortex (Yamada et al., 2021), periaqueductal grey (Lau et al., 2020), and hippocampus (Lu et al., 2021), opioids have been shown to exert presynaptic effects on glutamatergic transmission.”

We also now discuss in more detail the use of young animals in vitro vs adults in vivo, and how this approach is a major strength of our study. e.g. Line ~312:

“Thus, our finding that photoinhibition of Oprm1+ preBötC neurons did not phenocopy OIRD was consistent among in vitro and in vivo preparations, despite their inherent differences e.g. neonates vs. adults, level of extracellular [K+], temperature, and effects of anesthesia.”

4 – Attention should be put on statistics that are not clear enough in the present state (P values and tests should be given everywhere and must fit the illustrations, the text and the legends). Also Details on the number of animals used are missing. More importantly, an important issue is whether the authors have adequately considered hysteresis effects associated with opioid exposure. This is critical, because the findings of this study have implications for a serious public health issue. For instance, if the dose-response component of their study included randomization of opioid concentration presentation order, then the Authors can analyze their data to assess whether response to 300 nM DAMGO in a naïve animal matches responses to the same concentration in an animal previously exposed to lower dosages. In contrast, if dosages were always presented in the same order, then it would be informative if they carried out experiments in which 300 nM DAMGO was applied to naïve animals, and compared responses to responses they've already obtained in their dose-response studies. If responses are not significantly different, then this issue can be disregarded. This should be at least discussed.

We’ve updated all of the statistics, figures, and legends, and have included additional details where requested. Detailed statistical information and data are also now included in the Source Data files for all Figures and figure supplements.

We now include new data examining potential hysteresis effects (Figure 3—figure supplement 1). In our initial submission, DAMGO was always presented in the same order (50, 100, 200, 300 nM) and for the same duration (7min in each concentration), thus, differences between groups cannot be explained by hysteresis effects. Further, the added experiments demonstrate that the response of a “naïve” preBötC network to 200nM DAMGO is the same as the response of a network previously exposed to lower dosages of DAMGO, suggesting that development of tolerance or desensitization doesn’t play a significant role in the respiratory network on the time scale tested here (minutes) consistent with observations that development of respiratory tolerance to opioids may occur more slowly and via distinct mechanisms than in e.g. pain circuits (Hill et al., 2019; 2019; Withey et al., 2017).

Reviewer #1 (Recommendations for the authors):Baertsch et al., present a very compelling study investigating network level mechanisms underlying opioid-induced respiratory depression. This study is topical considering the respiratory side-effects of opioids and their substantial misuse in North America. The authors propose that MORs inhibit a subset of neurons expressing Oprm1 and that respiratory network inhibition is due to reduction of pre-inspiratory spiking and inhibition of excitatory synaptic transmission. The study is mechanistic, it is new and, while the experiments are technically challenging, they are overall well-executed. There are limitations to the study that preclude its publication in its current state. The major limitations are related to data analysis, presentation, and interpretation. The comments below aim to improve the study and can be addressed by the authors. Some controls such as histological analysis of opsin channel expression in Oprm1 neurons in the region of the preBötC may be necessary. Overall, there all a lot of data presented and it may be difficult for some readers to follow the discussion. The authors could simplify the results and discussion to make easier to understand for a general audience. However, I believe that all the comments listed below can be addressed by the authors.

We thank the reviewer for the thorough review and constructive comments!

1) The main conclusion of Figure 4 is that inhibition of Oprm1 cells in vitro and in vivo do not replicate the effects of DAMGO or morphine. Or that hyperpolarization of oprm1 cannot fully account for OIRD. Although I tend to agree with this conclusion, how can the authors be sure that all Oprm1+ cells expressed opsin channels, or that a sufficient amount of channels is expressed? One challenge with cre-lox recombination approaches is to demonstrate whether the cells expressing cre recombinase are expressing ArchT. If the Oprm1/cre expression is relatively low, then the opsin channels may not be highly expressed which would limit the inhibition by light. What is the expression of ArchT in Oprm1+ neurons?

This is a critical aspect of our study, and it is important to emphasize that our primary goal was not to determine the role of Oprm1+ neurons in the preBötC, but instead to examine the specific functional consequences of MOR activation in the network. These are two different questions that should be tackled using different approaches. For example, if the activity of all Oprm1+ neurons was completely silenced such that they could no longer contribute to preBötC function, the rhythm is likely to stop (as suggested by our modelling experiments shown in Figure 7—figure supplement 1). However, a key finding of our study is that this is not what occurs during OIRD since the spiking activity of Oprm1+ neurons is only moderately suppressed during severe OIRD (e.g. Figure 2 and 4C), and therefore this approach does not adequately address the underlying mechanisms. Thus, the strength of photoinhibition was carefully titrated to mimic the amount of network suppression induced by DAMGO in order to explore the role of membrane hyperpolarization in OIRD.

The reviewer is correct that Cre-Lox strategies are not without limitations. For instance, when using viral approaches, incomplete expression of opsin due to inefficient viral transduction and/or anatomical variation in injection sites can lead to an under-estimation of the functional role of the target cell population. However, when using an intersectional genetic approach like we did in this study, Cre-Lox recombination is highly efficient (e.g. Orban et al., 1992; Zheng et al., 2000), and only a low level of Cre expression is required for recombination. As a result, Cre-Lox optogenetic approaches often result in an amplification effect in the resulting opsin expression relative to the level of “driver gene” (Oprm1) expression. For example, even neurons with a low level of Oprm1 expression (and also little MOR immunoreactivity and a low sensitivity to MOR agonist) are likely to undergo Cre-dependent recombination. Once recombination occurs, the opsin (ArchT) is expressed under control of the Rosa26 gene– which is highly expressed, constitutive, and ubiquitous. Thus, the level of Oprm1 expression and resulting ArchT expression is not proportional. Therefore, our hyperpolarizing manipulations of Oprm1 neurons would be more likely to exaggerate rather than under-estimate the extent of network hyperpolarization induced by MOR activation. This would only further support our conclusion that MOR-induced hyperpolarization of preBötC Oprm1 neurons per se is not sufficient to mimic OIRD.

Supporting information from JAX regarding the Oprm1CreGFP mouse strain: “Oprm1 is normally not expressed abundantly, thus the GFP fluorescence is not directly visible and barely detectable with antibodies to GFP.” “This strain has been tested with adeno-associated virus expressing double-floxed inverse ORF fluorescent protein (AAV‐DIO‐FP; either GFP or mCherry) injected into a brain region where expression was expected and it revealed robust fluorescence.”

Is the expression of the opsin channels high enough to match the effects of MOR inhibition by DAMGO. The authors need to demonstrate that all neurons expressing Oprm1 are expressing ArchT at a sufficient level.

We specifically tailored the intensity of our light stimulations so that activation of ArchT did indeed mimic the effects of MOR inhibition by DAMGO. We did this based on two main factors:

1) We “matched” the change in integrated network spiking activity induced by 300nM DAMGO with the change in integrated spiking activity during photoinhibition of Oprm1 neurons. in vitro, 2mW was sufficient for this (top panel in Figure 4D), and 6mW was sufficient in vivo (top panel in Figure 4I). At these light intensities, photoinhibition reduced integrated network spiking activity by an amount equivalent to that observed in 300nM DAMGO, but did not cause a similar frequency depression. This supports our conclusion that hyperpolarizing Oprm1 neurons to mimic the change in spiking activity elicited by DAMGO is not sufficient to reproduce OIRD.

2) At the level of single neurons, we found that photoinhibition resulted in membrane hyperpolarizations (Figure 4—figure supplement 1) consistent with what has been previously shown in response to DAMGO (e.g. Montandon et al., 2011; Mellen et al., 2003). However, this is not necessarily straight forward since, as demonstrate in our modeled Oprm1+ neurons (Figure 7B), the amount any given neuron hyperpolarizes in response to a MOR-mediated current can vary significantly depending on its specific e.g. membrane resistance, capacitance, MOR expression, and other intrinsic properties. Thus, we preferred to use the network-wide changes in integrated spiking activity described above as our primary functional readout to “mimic” the hyperpolarizing effects of DAMGO.

Was there some staining or microscopy done to confirm this? These are critical control that need to be performed for all experiments using optogenetics.

We used a functional electrophysiological strategy to confirm that light sensitive (Cre-expressing) neurons were also sensitive to DAMGO, whereas neurons that were not light sensitive (not Cre-expressing) were much less sensitive to DAMGO (Figure 2D), providing strong evidence that opsin and MOR were expressed on the same preBötC neurons. This is now more clearly stated in the text. Further, the Oprm1Cre mouse line has been validated previously by the donating lab with whom we are in close collaboration (Liu et al., 2021, PNAS), and we also observed that Oprm1-driven opsin expression matched the expected anatomical expression patterns of Oprm1 mRNA that have been previously shown by the Allen institute using ISH (Figure 1—figure supplement 1). We initially considered the suggested immunohistochemistry experiments; however this approach has many of its own limitations e.g. antibody specificity, cross-reactivity, background staining, etc., and is not necessarily more specific than the highly efficient intersectional genetic approach described above. Therefore, because of its prior validation and the specificity indicated by our electrophysiological results, we did not perform immunohistochemistry to look for colocalization of MOR immunoreactivity and Cre-reporter expression.

2) Neonatal in vitro versus adult in vivo. The authors often compare experiments done in neonatal in-vitro preparations with adult in vivo experiments. The authors should mention clearly what are the potential differences between these experiments: immature respiratory network, differences in opioid sensitivity, raised potassium, room temperature versus adult network, anesthesia etc. This should be discussed.

We couldn’t agree more with the reviewer, and we think this is a major strength of our study since our results in vivo do support or findings in vitro despite all of the caveats and differences associated with these preparations.

In Results line ~312, we now clearly state:

“Thus, our finding that photoinhibition of Oprm1+ preBötC neurons did not phenocopy OIRD was consistent among in vitro and in vivo preparations, despite their inherent differences e.g. neonates vs. adults, level of extracellular [K+], temperature, and effects of anesthesia.”

3) Statistical comparisons. There are many experiments where some two-way ANOVAs were shown in figure legends, but the p-values of post-hoc tests are not presented.

In such cases, p-values of post-hoc tests were not significant. All significant and non-significant p-values for ANOVAS as well as the associated post-hoc tests can be found in the Source Data files.

A lot of percentages are presented without statistical tests. This need to be addressed. For instance Figure 4C. It is said in text that DAMGO has an effect on IBI but it is not shown with a * in Figure. P-values and n should be consistently presented in text or legends. This applies to all figures.

Additional statistical information has been added throughout the results and in the figure legends. However, there is a lot of statistical information associated with each figure panel and experiment. We feel that including all statistical details (n=, test performed, p-values for both ANOVAs and posthoc tests) in the results in particular, but also the legends would clutter the text and become confusing. All of this detailed information is clearly laid out in table format in the included source data files.

4) I would suggest to the authors to add to Figure 7 (or a separate Figure 8) a schematic of types of preBötC neurons affected by DAMGO. It may be difficult to the readers to conceptualize the impact of DAMGO on preBötC neurons without seeing a summary of different roles and responses of neurons. Could the authors present a table summarizing the impact of DAMGO on pre-inspiratory, inspiratory, oprm1- and oprm1- neurons? Or mechanisms. Or a diagram?

We have added a Figure 8 that summarizes the effects of MOR activation on different types of pre-inspiratory and inspiratory preBötC neurons.

5) Presentation of data and results. The presentation of the data could be improved. Indeed, the important results presented in many figures can be hard to identify. For instance, in figure 2, panel C is very small and not highlighted. If the authors could find a way to increase the size of panels, it would greatly improve their impacts. This applies to all figures.

We understand the reviewer’s point. But after going through many iterations of figure 2 and the rest of the figures, we feel that this configuration was the most effective way to display the data. Together, Figures 2C and 2D show clearly that DAMGO suppresses spiking of Oprm1+, but not Oprm1-, pre-inspiratory neurons during the inter-burst interval, and we have emphasized the importance of this finding in the text.

6) The mechanisms identified here should be compared to existing work related to opioid-mediated inhibition for other model systems such as MOR inhibition in pain circuits.

We agree and have revised the text accordingly. E.g. we now state in RESULTS:

“The mechanisms of opioid action are diverse and vary based on brain region (Crain and Shen, 1990, Yaksh, 1997, Bourgoin et al., 1994, Christie, 1991). However, in many neuronal circuits including the parabrachial nucleus (Cramer et al., 2021) prefrontal cortex (Yamada et al., 2021), periaqueductal grey (Lau et al., 2020), and hippocampus (Lu et al., 2021), opioids have been shown to exert presynaptic effects on glutamatergic transmission.” And we state in DISCUSSION “MOR activation also weakened excitatory synaptic connections in the preBötC (Figure 5D, E), similar to previous observations in spinal nociceptive pre-synaptic terminals (Corder et al., 2017)….”

7) The authors should comment and discuss the fact that opioid drugs such as morphine and fentanyl may have different impacts in mice compared to humans, and that dose-dependent effects are likely observed as previously shown by other authors (Albert Dahan etc.).

We state in the introduction:

“Although studying the underlying mechanisms of OIRD in humans remains difficult, in both humans and mice, OIRD is characterized by a pronounced decrease in the frequency and regularity of breaths (Bouillon et al., 2003, Ferguson and Drummond, 2006, Smart et al., 2000). This is primarily due to longer and more irregular pauses between inspiratory efforts (Drummond, 1983).”

Dose dependent effects were examined in our study by increasing [DAMGO] in a stepwise fashion from 50-300 nM.

Reviewer #2 (Recommendations for the authors):1. Although the number of cells recorded from is documented, the number of animals used is not mentioned anywhere in the manuscript. How many opioid exposures were carried out in each experiment? Was the order of presentation of different DAMGO concentrations varied? What was the interval between DAMGO exposures?

We thank the reviewer for these important points. We now include all of this information in the results, figure legends, and/or methods. In figure 1, n=223 neurons were recorded from n=73 horizontal slices. In all of the other intracellular recording experiments/figures, only a single neuron was recorded from each slice preparation, i.e the number of animals used is the same as the number of cells recorded. This was done specifically so that every cell was “naïve” to opioid exposure at the time of recording. We now specifically state this information in the text, line ~191:

“A single neuron was recoded from each slice preparation and all were naïve to opioids at the time of exposure.”

For each experiment (i.e. figure panel and statistical comparison), all slices/animals experienced the same number of opioid exposures in the same order. For example, the results from slice preparations in Figures 2, 3, 4, and 6 are repeated measures data during sequential step-wise application of DAMGO at 50, 100, 200, and 300 nM DAMGO (7min in each condition with data analyzed from the last ~2min of each condition). This is mentioned in METHODS. Because all groups were treated in the same order for the same duration, differences between groups cannot be attributed to e.g. MOR desensitization or development of tolerance. Although this was not a focus of our study, it is interesting to note that desensitization and tolerance to OIRD seems to be substantially less robust and slower to develop compared to opioid tolerance in other contexts e.g. pain– and seems to involve distinct mechanisms (Hill et al., 2019; 2019; Withey et al., 2017). Based on our results, one could speculate that these differences reflect the relative functional consequences of the intrinsic vs synaptic effects of opioids on the circuit.

2. 4. The Butera model of respiratory rhythmogenic networks is computationally tractable, high-level, and low-dimensional. Thus, it is appealing because it helps us develop intuitions about this rhythmogenic network. In the original model, pacemaker neurons were obligatory constituents. At the time the model was developed, this was unproblematic, because there was broad consensus that respiratory rhythm arose out of the activity of a kernel of endogenously bursting (EB) pacemaker neurons. As anyone who has followed the literature on this topic knows, the existence and significance of EBs within the PBC has become contentious, with both their relative number and functional significance brought into question.It is puzzling that the Authors didn't implement their in silico simulation based on a more recent model that does not require the presence of EBs, and maintains stable rhythmicity via synaptic interactions between pre-I neurons (Guerrier et al., 2015). This model is an attractive alternative, because phasic variation in synaptic efficacy is the mechanism for rhythmogenesis. Thus, the dual action of opioids on presynaptic excitability and post-synaptic efficacy proposed here as the mechanism for OIRD could straight-forwardly be implemented.

The reviewer brings up a good point regarding the role of endogenous bursting neurons. The model we present here includes INaP, but none of the neurons have intrinsic bursting activity when synaptic interactions are blocked. All neurons are either intrinsically tonically firing (35%) or quiescent (65%). We did this because (1) as the reviewer mentions, the “critical” significance of endogenous bursters has been questioned, (2) in this study, all neurons we found in our in vitro patch experiments were either tonic or quiescent under baseline conditions in CNQX/CPP, and (3) the model generates regular bursting without an endogenously bursting subpopulation included, and adding ~5-10% endogenous bursting neurons in the model didn’t significantly impact our results or conclusions.

In contrast to the specific role of endogenous bursting neurons, the presence of INaP within preBötC neurons is well-supported (e.g. Del Negro et al., 2002; Koizumi and Smith 2018) and generally accepted. The Guerrier et al., 2015 model is important because it supports the hypothesis that intrinsic burst-promoting currents (e.g. INaP) may not be required for rhythmogenesis under all conditions, and that synaptic mechanisms per se may be sufficient given certain dynamics. However, this does not mean that INaP doesn’t exist or play an important functional role within the preBötC. Thus, excluding INaP from our computational experiments did not seem appropriate to address our research question, and instead we used an adapted version of the tractable and widely implemented Butera model in order to examine the potential interaction of hyperpolarization and/or presynaptic suppression on the network-level activity.

Reviewer #3 (Recommendations for the authors):This study investigates the important problem of network- and cellular-level mechanisms of opioid-induced respiratory depression (OIRD) operating within the preBötzinger complex inspiratory rhythm generator, which has been shown to be one of the critical brainstem respiratory regions involved. The authors present a detailed electrophysiological analysis in mice of the spiking patterns of preBötC neurons expressing mu-opioid receptors (MOR) and examined how spiking patterns and preBötC population activity is affected by MOR agonists in vitro and in vivo. The studies incorporated optogenetic approaches to identify by photostimulation neurons expressing the MOR encoding gene Oprm, and to manipulate activity of these neurons by photoinhibition. The authors present important data on the electrophysiological phenotypes of Oprm+ neurons. They present novel evidence that MOR activation reduces inspiratory neuron spiking particularly of neurons with pre-inspiratory spiking patterns and depresses excitatory synaptic transmission- dual mechanisms of opioid action that the authors propose can account for the perturbations of inspiratory rhythm generation. Simulations with a preBötC network biophysical model also suggest that these dual mechanisms in combination are required to explain how inspiratory rhythm generation can be disrupted by MOR activation. The paper is well generally well written and the data presented for the most part advances understanding of the mechanisms of OIRD at the level of preBötC circuits.

We thank the reviewer for their constructive review.

1) The authors are proposing that MOR activation-induced hyperpolarization of preBötC excitatory neurons alone is not sufficient to explain disruption of inspiratory rhythm generation. This is based largely on the experimental observations that optogenetic hyperpolarization of Oprm+ preBötC neurons does not produce the same level of depression of preBötC neuronal spiking activity and inspiratory burst frequency as DAMGO. The authors do not present data showing that the depolarization produced by photoinhibition is comparable to that produced by DAMGO. The only figure that could potentially illustrate effects of DAMGO on membrane potential of inspiratory neurons from whole-cell recording is Supplemental Figure 4, which is a nice and necessary validation of their transgenic mouse construct, but this figure does not indicate values of baseline membrane potentials before and after DAMGO. The authors should provide more detailed information on the level of membrane hyperpolarization produced by photoinhibition compared to that produced by DAMGO.

Hyperpolarization of preBötC neurons in response to MOR activation has been shown previously (e.g. Montandon et al., 2011; Mellen et al., 2003), which is generally consistent with the amount of hyperpolarization we induced during photoinhibition. However, there is considerable variability in the magnitude of the hyperpolarizing response to MOR activation between research studies and between individual neurons (Ballanyi et al., 2010; Mellen et al., 2003). This is not surprising given our modelling results (Figure 7B), demonstrating that the amount any given neuron hyperpolarizes in response to a MOR-mediated current can vary significantly depending on its specific intrinsic properties e.g. membrane resistance, capacitance, gNaP, etc. Differences in the abundance of MOR expression between neurons to is also likely to contribute to variable responses. Therefore, instead of trying to match MOR induced hyperpolarization at the level of single neurons, we used a network-level approach and “matched” the change in integrated network spiking activity induced by 300nM DAMGO with the change in integrated spiking activity during photoinhibition of Oprm1 neurons. in vitro, 2mW was sufficient for this (top panel in Figure 4D), and 6mW was sufficient in vivo (top panel in Figure 4I). At these light intensities, photoinhibition reduced integrated network spiking activity by an amount equivalent to that observed in 300nM DAMGO but did not cause a similar frequency depression. Thus, supporting our conclusion that hyperpolarization alone in not sufficient to mimic OIRD in the preBötC. We have revised the text to clarify this.

2) The authors present interesting modeling results suggesting that hyperpolarizing model neurons incorporating opioid-induced currents vs. reducing their synaptic output has effects on network spiking activity resembling the differential effects on cellular/network activity observed during optogenetic hyperpolarization of Oprm+ neurons vs. MOR activation. There is no discussion of the modeling results in terms of the model biophysical mechanisms and formulation. This model in general can show that hyperpolarization of key rhythmogenic neurons (e.g., via K^+^-dominated leak currents) can produce the perturbations of the rhythm similar to MOR activation, depending on parameterization of the conductances. The authors present an interesting formulation of the Butera et al. model in terms of the range of NaP conductances and the values of excitatory synaptic conductances, both of which affect the rhythmogenic capabilities of this model. Why was the model parameterized as specified?

The text has been modified at e.g. lines ~415-425 to justify the use of the Butera model and the parameters chosen. The Butera model is chosen as it is low-dimensional, simple and effective in producing regular bursting behaviors at the network level, thus allowing for examination of the network level effects of implementing either, or both, neuronal hyperpolarization and synaptic depression. The specific parameters were modified from previously published work that incorporated inhibitory populations (Harris et al., 2017). Inhibitory populations are a key component of the respiratory network, but not included in many published models (including the Guerrier model). Parameters were taken from Harris et al., 2017 but NaP and leak conductances were modified to better qualitatively match in-vitro observations of burst frequency and duration. Further, the model was formulated such that it contained no intrinsically bursting cells (i.e., cells do not exhibit bursting behavior in the absence of synaptic connections). We did this because (1) the “critical” significance of endogenous bursters is controversial, (2) in this study, all neurons we found in our in vitro patch experiments were either intrinsically tonic or quiescent (not bursting) (3) the model generated regular bursting without an endogenously bursting subpopulation included, and adding ~5-10% endogenous bursting neurons in the model didn’t significantly impact our results or conclusions.

Why was the paper of Bacak et al., (eLife 2016;5:e13403. DOI: 10.7554/eLife.13403) not referenced in terms of how cellular activity synchronization due to synaptic connection weights influences the population activity amplitude fluctuations and other aspects of how the basic model employed here produces circuit dynamics relevant to this study?

Thanks for bringing this to our attention. We now include references to the important study by Bacak et al., which is consistent with our finding of an increased probability of burst failures in DAMGO. Specifically, we now state:

“Model networks also exhibited failed bursts reminiscent of our findings in vitro (see Figure 3D), which may represent the emergence of mixed-mode oscillations as previously shown in heterogeneous networks with sparse synaptic connectivity (Bacak et al., 2016).”